# An error correction strategy for image reconstruction by DNA sequencing microscopy

Alexander Kloosterman, Igor Baars & Björn Högberg ✉

By pairing adjacent molecules in situ and then mapping these pairs, DNA microscopy could substantially reduce the workload in spatial omics methods by directly inferring geometry from sequencing data alone. However, experimental artifacts can lead to errors in the adjacency data, which distort the spatial reconstruction. Here we describe a method to correct two such errors: spurious crosslinks formed between any two nodes, and fused nodes that are formed out of multiple molecules. We build on the principle that spatially close molecules should be connected and show that these errors violate this principle, allowing for their detection and correction. Our method corrects errors in simulated data, even in the presence of up to 20% errors, and proves to be more efficient at removing errors from experimental data than a read count filter. Integrating this method in DNA microscopy will substantially improve the accuracy of spatial reconstructions with lower data loss.

With the improvements in sequencing technology, techniques to investigate biological samples have become increasingly refined, progressing from sequencing in bulk, to single-cell RNA[1] and spatial transcriptomics[2,3]. The latter technique allows one to obtain the organization of cells in tissue, leading to deeper insights in biology and improving the detection of diseases[4,5]. However, current techniques for spatial transcriptomics rely on fluorescence microscopy, which is limited in throughput, especially for large amounts of targets[3].

DNA microscopy is an emerging spatial transcriptomic technique that aims to find the spatial organization of DNA or RNA using sequencing alone, bypassing the use of optical microscopy. The common theme in all DNA microscopy methods is to use a polymerase chain reaction (PCR) and sequencing to find pairs of adjacent molecules and use that pairing information to find their relative locations[6–10] (Fig. 1). In a typical workflow (Fig. 1a), molecules are barcoded and amplified locally in the tissue of interest, creating polymerase colonies (polonies)[11] (Fig. 1b). Where polonies overlap, amplicons fuse by design and form concatemers (Fig. 1c), which, when sequenced, reveal which two polonies are adjacent. All the adjacency data are represented in a graph, where each node represents one polony, and each edge a sequenced concatemer. From this graph, the original locations can be estimated[6,7,9,10] (Fig. 1d).

Importantly, the information of these adjacency pairs is obtained with sequencing information only, meaning that (1) the method can capture both the sequence and location of transcripts simultaneously, and many targets can be captured simultaneously, (2) it does not require processing or stitching of image data, only analysis of sequencing data, and (3) it is not inherently limited to two-dimensional (2D) reconstructions, but can be used to reconstruct 3D samples as well.

However, experimental conditions can give rise to erroneous signals that create artifacts in the adjacency graph and disrupt the spatial reconstruction. We consider two such errors. The first of these is that of spurious crosslinks, formed between any pair of nodes regardless of position. These can be formed by incomplete PCR during post-experimental library preparation when products are no longer spatially confined, in a reaction similar to barcode-swapping and index-hopping events[12]. The second type of error is a fused node. When two polonies contain the same barcode or very similar barcodes that are mistakenly fused by sequencing error correction, they are represented in the adjacency graph as a single node, which can lead to distortions in the reconstruction.

In this Article we propose two methods to remove these errors, collectively called MinIPath ('Minimum Indirect Path' analysis). These methods are based on graph analysis on the adjacency graph alone.

Department of Medical Biochemistry and Biophysics, Karolinska Institutet, Stockholm, Sweden. ✉e-mail: bjorn.hogberg@ki.se

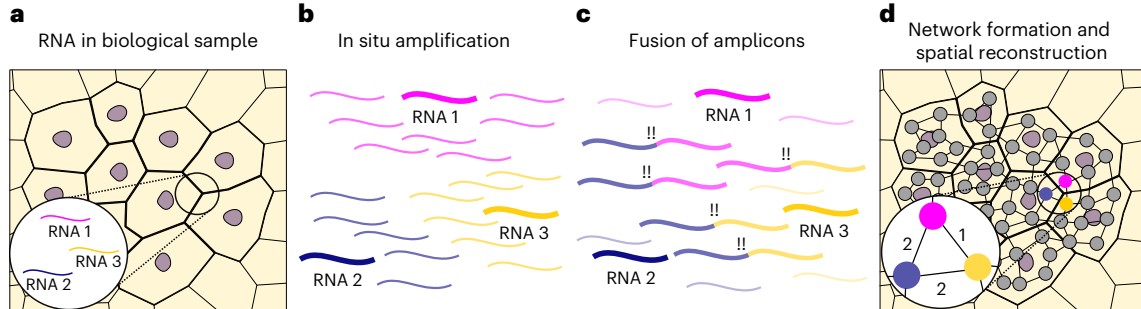

**Fig. 1 | DNA microscopy reveals spatial locations by finding adjacent pairs of transcripts. a**, RNA is present in a biological sample of interest. **b**, RNA molecules are barcoded and amplified locally, forming polymerase colonies (polonies). **c**, Where polonies overlap, their amplicons can be engineered to fuse together, forming concatemers. **d**, Sequencing concatemers reveals the adjacency of all polonies, which can be displayed in a graph. The number of concatemers formed between two polonies is the edge weight. From this graph, the relative spatial coordinates of each molecule can then be obtained.

Spurious crosslinks are detected by finding a short indirect path connecting two directly connected nodes, which we show is harder to find if two nodes are far away and erroneously connected. Fused nodes are detected by looking at the connected nodes of any node. When these can easily be separated into two indirectly connected groups, the node is probably a fused node and can be split. We show the effect of both types of error on the reconstruction quality using simulated diffusion-based data as input, and that these can be corrected by our method. In addition, we analyze a previously described DNA microscopy dataset[7] and show that we can obtain accurate reconstructions by removing spurious crosslinks more efficiently than with a read count filter. In summary, this method provides an efficient way to filter artifacts from adjacency-based data, which can improve the overall quality of the resulting spatial reconstruction.

## Results

### Correcting errors in simulated data using graph organization

Although the principle of imaging space by sequencing can be realized in various experimental set-ups, we focus here on a set-up introduced previously[7] that has yielded experimental results. In this arrangement, the sample is encapsulated in a hydrogel, meaning that polony formation is governed by diffusion. Two types of seed strand are present in the sample, to prevent the self-interactions of polonies, which could consume large amounts of sequencing data without providing information on neighbors. Furthermore, each formed concatemer contains a unique barcode, allowing one to count the number of interactions between two polonies. All the neighborhood interactions are represented in a neighborhood graph, where the weight of the edges equals the number of observed products between two polonies. As two types of seed strand are used, the neighborhood graph is a weighted, undirected, bipartite graph.

We first sought to simulate this experimental set-up. Starting from Fick's law for diffusion for a single polony, one can derive the relationship between the reaction rate $\omega$ between two polonies and their distance[7]:

$$\omega(i, j) \propto t^{-\frac{d}{2}} \times \exp\left(\frac{-|\mathbf{x}_i - \mathbf{x}_j|^2}{L_{\text{diff}}^2}\right) \qquad (1)$$

$$L_{\text{diff}} = \sqrt{8dDt} \qquad (2)$$

where $\omega(i, j)$ is the reaction rate between two polonies $i$ and $j$, each of a different type, $D$ is the diffusion constant, $t$ is the time since polony creation, $d$ is the number of dimensions, and $x_i$ and $x_j$ are the locations of polonies $i$ and $j$, respectively. $L_{\text{diff}}$ describes the characteristic diffusion length of the distribution.

As polony input, we randomly distributed nodes of two types within a 2D shape (Fig. 2a; total dimensions 200 × 450 pixels; shape area ~40,000 pixels; ~4,000 nodes). The diffusion model was used to estimate the number of reactions between each pair of nodes, and this

number was used as a parameter in the Poisson distribution to obtain connections and their edge weights:

$$w(i, j) = \text{Poiss}\left(a \times \exp\left(\frac{-|\mathbf{x}_i - \mathbf{x}_j|^2}{\sigma^2}\right)\right) \qquad (3)$$

where $\omega(i, j)$ represents the edge weight between two nodes $i$ and $j$, $a$ represents the amplitude and $\sigma$ the spread. In experimental terms, the amplitude can be affected by the inherent reactivity of the polonies and sequencing depth, both of which determine how many products are seen. If the amplitude is high, multiple concatemers are formed and/or sequenced for each polony pair, resulting in larger edge weights in the resulting graph. The spread is analogous to $L_{\text{diff}}$ and determines the distance at which two polonies might still react. Experimentally, it will be determined by the diffusion, and therefore by the properties of the hydrogel and size of the products.

As a baseline, we generated adjacency data using a wide range of amplitudes and spreads (Fig. 2b) and reconstructed the polony locations using the previously described spectral maximum likelihood embedding (sMLE) method[7]. We reconstructed all adjacency datasets where at least 80% of all nodes were connected in a single group, and evaluated these reconstructions using two metrics: the Procrustes disparity as a global metric, and the number of overlapping neighbors out of the 15 nearest as a local metric[9].

The different parameters used in the simulation had a great influence on the reconstruction quality (Fig. 2c). Globally accurate reconstructions were obtained for many of the simulations. Only when a very low spread ($\sigma = 10$) or a combination of high spread and amplitude was used ($\sigma = 200$, $a = 100$) did the reconstructions become less accurate based on the global metric. Local accuracy depended primarily on the spread: if it was small ($\sigma \leq 50$), local accuracy was high even when global accuracy was low (k-nearest neighbors (KNN) $\geq 0.75$; Fig. 2d), but when it started to approach the sample size, local accuracy decreased (Fig. 2f,g). Higher amplitudes also led to better local reconstructions, and the combination of low spread and high amplitude led to the most accurate reconstructions both globally and locally (Fig. 2e).

We then developed algorithms to detect and correct spurious crosslinks and fused nodes (Fig. 3). As mentioned above, spurious crosslinks randomly connect any two nodes of different types, regardless of the position of their corresponding polonies in the original sample (Fig. 3a). Similarly, node fusion results in nodes that inherit the connections of two randomly selected nodes, again regardless of the position of their corresponding polonies in the sample (Fig. 3b).

For spurious crosslinks, we reasoned that polonies are usually surrounded by many other polonies to which a connection is possible. This implies that when two nodes are connected in a graph, it should be straightforward to find a short, indirect path in the graph to connect these two nodes, without using the edge itself (Fig. 3c, left).

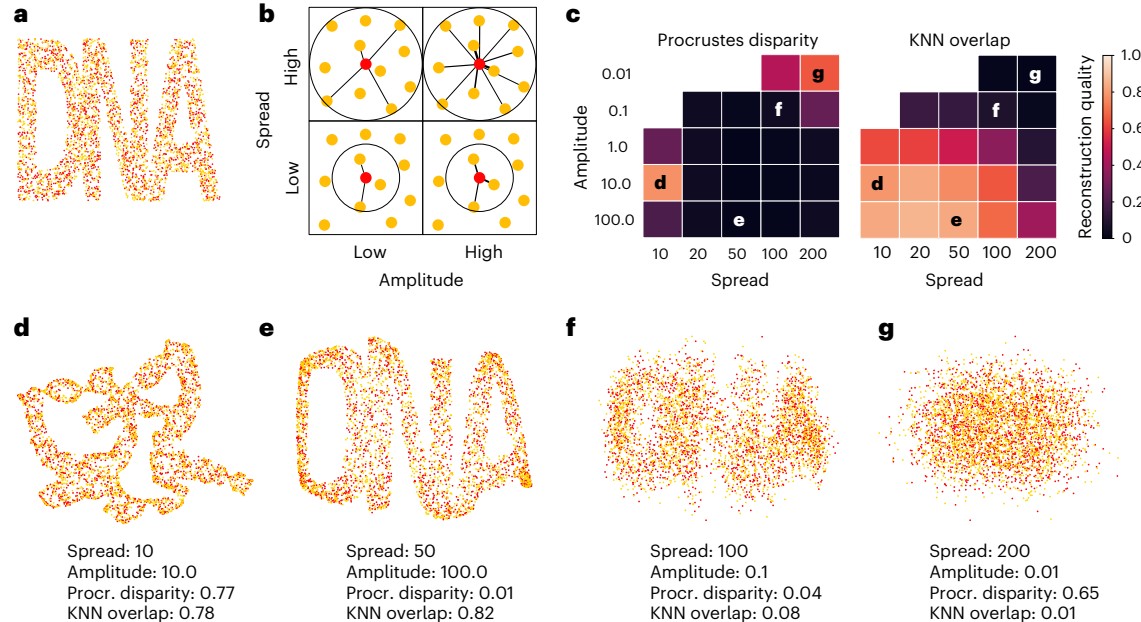

Spread: 10
Amplitude: 10.0
Procr. disparity: 0.77
KNN overlap: 0.78

Spread: 50
Amplitude: 100.0
Procr. disparity: 0.01
KNN overlap: 0.82

Spread: 100
Amplitude: 0.1
Procr. disparity: 0.04
KNN overlap: 0.08

Spread: 200
Amplitude: 0.01
Procr. disparity: 0.65
KNN overlap: 0.01

**Fig. 2 | Reconstructions quality depends on Gaussian parameters that determine adjacency. a**, Node locations used as input, on a grid of 200 × 450 pixels. **b**, Edges are formed between neighboring nodes. The spread determines the range at which edges are formed, and the amplitude determines the edge weight. **c**, The reconstruction quality depends on the amplitude and spread. **d**–**g**, Example reconstructions with different spreads and amplitudes: good local, poor global quality (**d**), good local, good global quality (**e**), poor local, good global quality (**f**) and poor local, poor global quality (**g**).

By contrast, spurious crosslinks are formed between polonies regardless of their distance. The further they are apart, the less likely it will be that a short, indirect path between the two nodes can be found (Fig. 3c, right). Using edge weights further amplifies the difference, as longer distance between nodes results in lower edge weights (equation (3)). To exploit this difference, for each edge, we find all indirect paths of length three (the minimum in a bipartite graph), calculate the product of the edge weights of each path and sum these together, to calculate what we will refer to as the indirect path value of that edge. We then use this to distinguish between normally connected nodes and spuriously connected nodes, removing the edge if it is below a certain cutoff.

For fused nodes, we built on the same reasoning that polonies are typically surrounded by other polonies to which a connection is possible. The nodes connected to any single, unaltered node should therefore also be connected to each other, with short, indirect paths, and form a single well-connected subgraph (using indirect paths of length two, the minimum in a bipartite graph; Fig. 3d, top). For the connected nodes of fused nodes, however, this is not necessarily the case. If the two polonies that are represented by a single fused node were sufficiently far apart in the sample, the connected nodes of the fused node should form two well-connected subgraphs, with many connections within each subgraph, but few connections between them (Fig. 3d, bottom). The original groups should therefore be obtainable with spectral graph partitioning[13], an algorithm that seeks to obtain graph partitions by minimizing the number of edges removed between them, while maximizing the number of edges within each partition. The ratio between these is called the normalized cut (ncut)[14]. The connected nodes of unaltered nodes can of course also be partitioned in two, but this requires the removal of many more edges, resulting in a higher normalized cut. The normalized cut can therefore serve to distinguish between fused and unaltered nodes. When it is below a given cutoff, it is replaced by two nodes, each inheriting the connections to the nodes in either of the graph partitions (Methods section Algorithm description).

To examine whether these algorithms proved effective, we first either added spurious crosslinks (1–20% of total edge weights) or fused nodes (1–20% of all nodes) in the simulated data. Adding in these errors distorted the reconstructions (for example, Fig. 4a). The global reconstruction quality was affected more than the local reconstruction quality, which can be understood as a small number of errors that twist the reconstruction without affecting the nearby neighbors of each point. We found that using a large spread when simulating connections made the resulting reconstructions more robust to errors, while the amplitude used in the simulation had little effect (Extended Data Fig. 1). When using a large spread, adding in spurious crosslinks is more likely to connect two nodes that are already connected, or fusing two nodes whose connected nodes broadly overlap, explaining the increased robustness to errors. Furthermore, because fusing nodes introduces nodes with twice as many connections, we verified that the resulting long-tailed distribution of node connectivity itself was appropriately normalized by the reconstruction pipeline and did not influence the reconstruction. To this end, we randomly introduced an increased reactivity bias to randomly selected nodes (1–20%) and found that this did not affect the reconstruction quality (Extended Data Fig. 1).

We applied our algorithm to the simulated data to calculate all indirect path values and normalized cuts, obtaining distributions of each for each simulated case (for example, Fig. 4b,e). On average, the spurious crosslinks connected nodes at larger distances from each other than normal connections, although there was some overlap (154.43 ± 90.43 for spurious crosslinks compared to 111.36 ± 74.91 for normal edges; average taken across all simulations). Still, the indirect path values were lower for spurious crosslinks compared to normal connections, with the exact values depending on the amplitude and spread used in the simulation, as well as the number of other errors (Extended Data Fig. 2). Similarly, the normalized cut values were lower for fused nodes (0.47 ± 0.33 for fused nodes compared to 0.71 ± 0.17 for normal nodes; average taken across all simulations). Again, distance played an important role here to detect each of the errors, as the crosslinks formed at longer distances had lower indirect path values, and fused nodes whose constituents were originally at a large distance could be partitioned with lower normalized cuts (Extended Data Fig. 2).

We then applied a range of cutoffs by taking the lower quantiles of all indirect path values or normalized cuts. Edges below their cutoffs

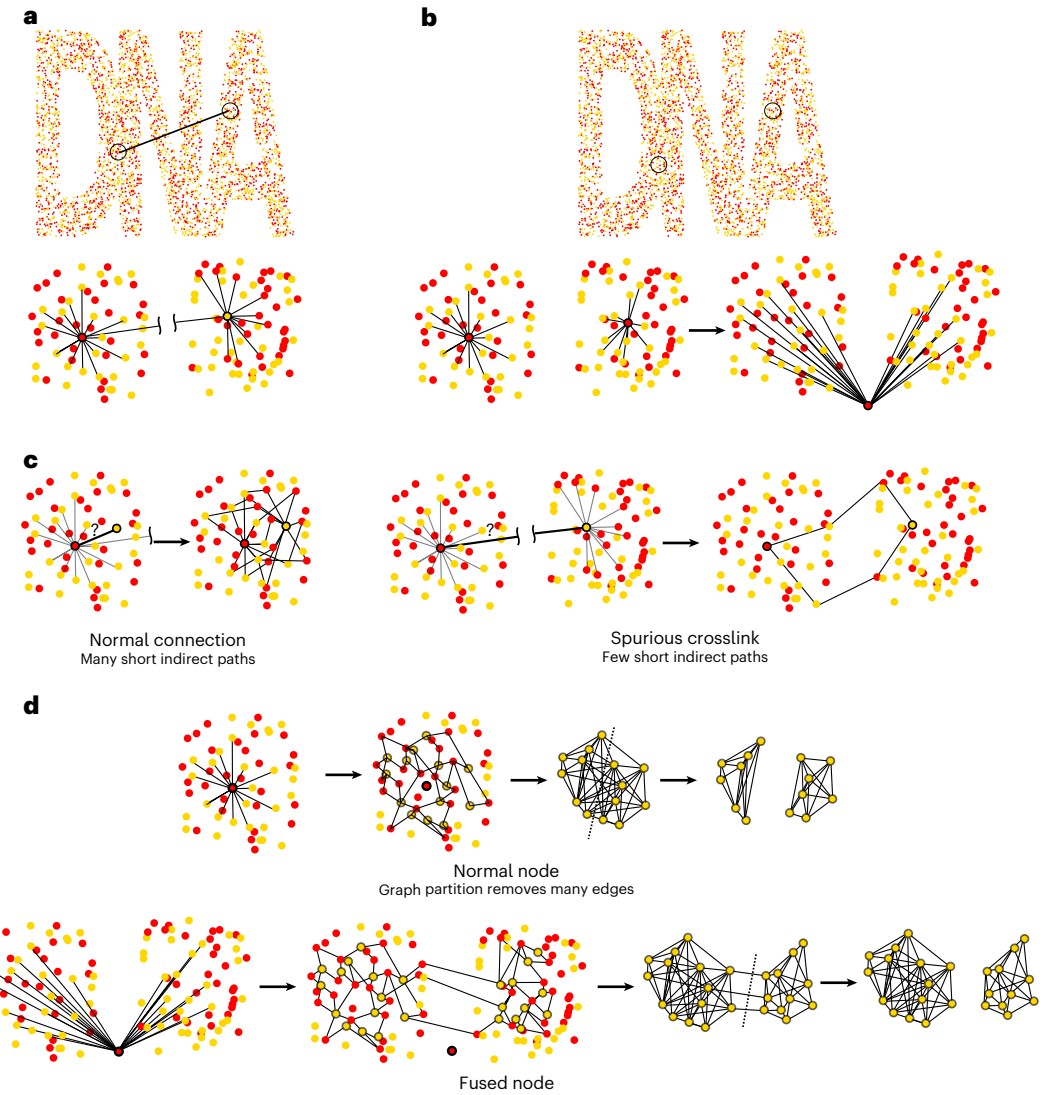

**Fig. 3 | Spurious crosslinks and fused nodes can be detected by indirect path analysis. a**, Spurious crosslinks, which connect two nodes of different types, regardless of distance. **b**, Fused nodes, in which two nodes are fused into one node, which inherits both of their edges. **c**, Spurious crosslink removal: obtain indirect paths of length three. For normal, local connections, one can find many short indirect paths connecting the same two nodes. However, when two nodes are spuriously connected and are sufficiently far apart, fewer short indirect paths will be found between them. **d**, Fused node splitting: separate connected nodes into two groups: normal nodes and fused nodes. A normal node is connected to several other nodes, which form a well-connected subgraph using short, indirect paths. Partitioning this subgraph removes many edges. However, the nodes connected to a fused node form two well-connected subgraphs, with only a few connections between them. Partitioning this graph therefore requires the removal of fewer edges.

were removed, and nodes with normalized cuts below their cutoff were split. Although not all errors could be removed, the correction algorithms preferentially corrected the spurious crosslinks and fused nodes over original edges and unaltered nodes on average across all simulations (Fig. 4g–i). The ratio of correctly identified errors (true positive) to original data identified as errors (false positive) depended primarily on the spread, that is, errors were more easily identified when only local reactions were formed (Extended Data Fig. 3). Notably, these reconstructions were also the ones that were most affected by the introduced errors in the first place (Extended Data Fig. 1). In addition, applying the algorithm without the presence of errors did not affect the reconstructions, except when an exceptionally high cutoff (quantile of 0.50) was used, suggesting that the error correction algorithm can safely be used on unaltered data, even at slightly too high cutoffs, without affecting reconstruction quality.

Applying the correction algorithm improved the reconstruction quality of any reconstruction that was affected by the errors

(Extended Data Figs. 4–7). Zooming in on simulations that were initially accurate but strongly affected by the errors (20 ≤ spread ≤ 100, amplitude ≥ 1.0), the average Procrustes disparity increased from 0.02 ± 0.03 to 0.63 ± 0.10 and to 0.52 ± 0.12, when spurious crosslinks or fused nodes were added, respectively. However, the average quality improved by applying the correction algorithm (Fig. 4k–n), and for each case where the introduced errors affected the reconstruction, a cutoff could be found that restored it (Extended Data Figs. 4–7). For most cases, this quantile cutoff was equal to the fraction of introduced errors, although when a higher cutoff was used, the reconstruction quality did not decrease. Using the expected fraction of errors in the data as a quantile cutoff therefore seems an appropriate guideline.

When fused nodes were corrected, we also evaluated the node splitting accuracy, that is, whether the nodes after splitting were connected to the same nodes as before the nodes were fused (Methods, Simulated data processing). We found that the resulting groups matched accurately to the original (average Jaccard index: 0.73 ± 0.17),

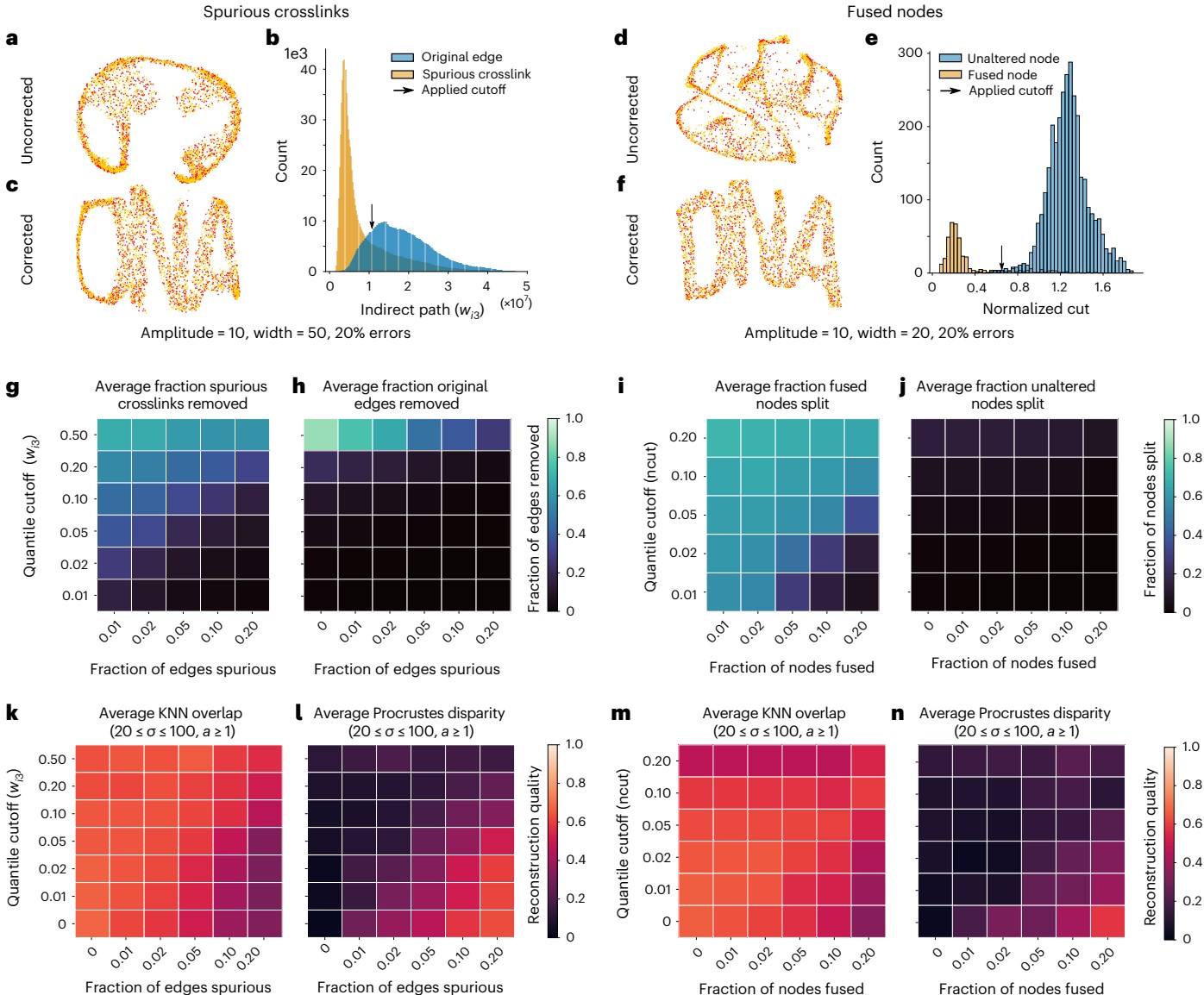

**Fig. 4 | Indirect path analysis rescues reconstructions by removing spurious crosslinks and correcting fused nodes. a,d,** Example distorted reconstructions at amplitude 10, and width 50 (**a**) and 20 (**d**), with 20% errors. **b,e,** Indirect paths and normalized cuts distributions of the pairing data used to generate the reconstructions in **a** (**b**) and **d** (**e**). The arrow indicates the cutoff used. **c,f,** Reconstructions after correction, corresponding to **a** and **d**. **g–j,** Average fraction of true positive (**g,i**) and false positive (**h,j**) across all simulated datasets. **k–n,** Average local (**k,m**) and global (**l,n**) reconstruction qualities before and after corrections across simulated datasets that were affected by the errors (with amplitude of 1, 10 or 100 and width of 20, 50 or 100).

and most accurately when the spread in the simulation was low (Extended Data Fig. 8). Node splitting only proved ineffective when spread and amplitude were low. In these cases, nodes were typically connected to only a few other nodes, which themselves were not indirectly connected to each other. The resulting indirectly connected graph was therefore often disconnected into multiple components, similarly to when a node is fused, making it impossible to identify whether the graph is easily partitioned due to sparse data or due to a fused node (Extended Data Fig. 8).

In summary, the proposed method removes spurious crosslinks and corrects fused nodes across a wide range of simulated data, even in the presence of up to 20% spurious crosslinks of 20% fused nodes.

**Correcting disruptive crosslinks in experimental data**
To see how our method would perform on experimental data, we analyzed a previously published DNA microscopy dataset for which a reference image was available[7]. In this experimental set-up, specific

types of RNA transcript (*ACTB* for 'beacons', and *gfp*, *rfp* and *gapdh* as 'targets') were used as seeds for the two types of polony in the bipartite graph. Each product connecting two polonies could be recognized by a unique barcode called a unique event identifier (UEI), the number of which was used as the edge weight between the respective nodes.

Using the same pipeline as described in the previous work[7], we extracted $1.26 \times 10^5$ polonies with $6.72 \times 10^5$ edges and $9.55 \times 10^5$ unique UEIs from the raw sequencing data. When using all data to reconstruct the largest connected group, the resulting reconstructions frequently collapsed into a 'star-like' pattern (Extended Data Fig. 9). Only one out of ten reconstructions produced a layout that could be overlaid on the microscopy image with a poor match (Fig. 5b). To remove possible artifacts, Weinstein et al.[7] applied a read count filter that removed all products without sufficient reads. Although this strategy did improve the reconstruction quality, a read count filter of four was required to obtain an accurate reconstruction (Fig. 5c,e). Given the large number of products produced during a DNA

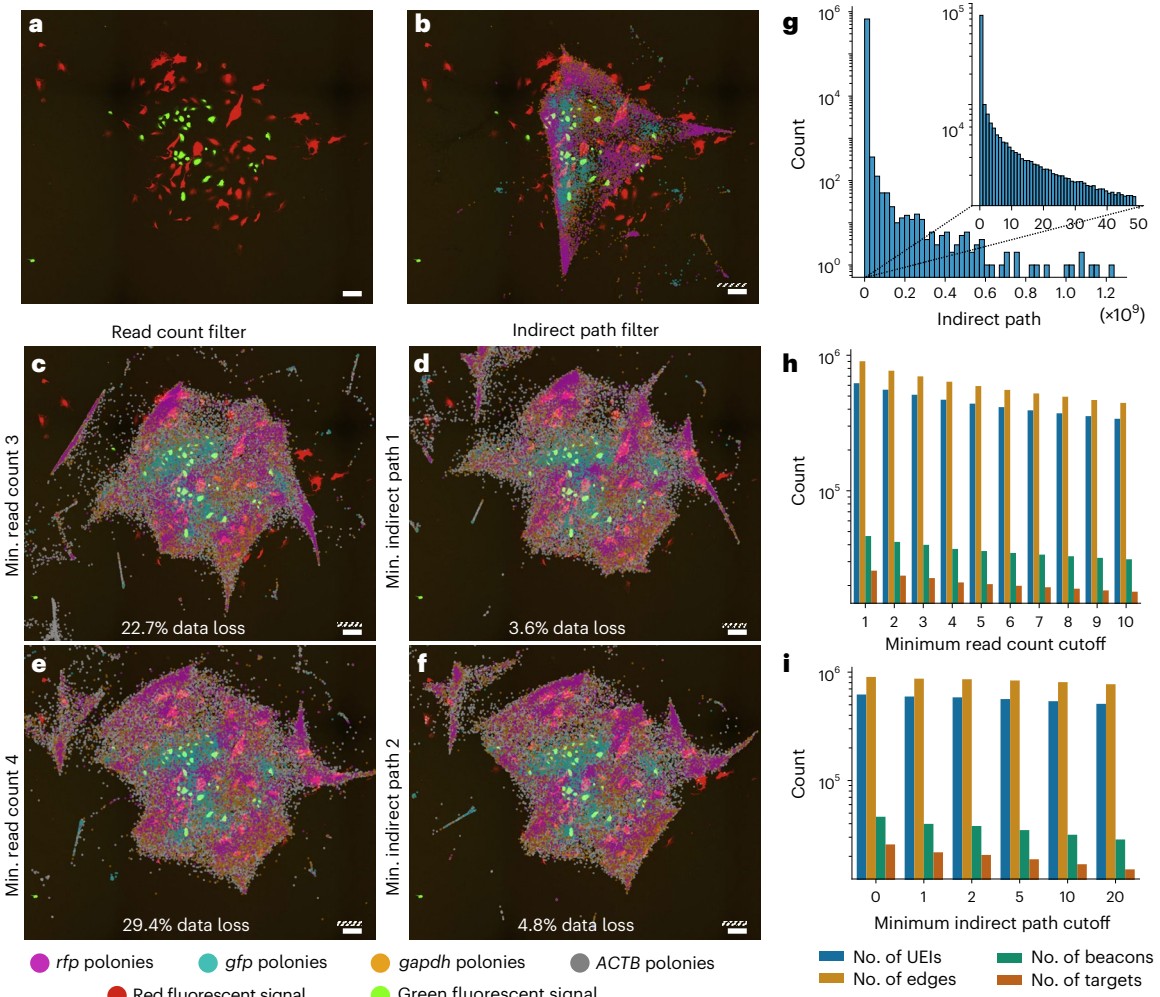

Read count filter — Indirect path filter

Min. read count 3 — 22.7% data loss

Min. indirect path 1 — 3.6% data loss

Min. read count 4 — 29.4% data loss

Min. indirect path 2 — 4.8% data loss

● *rfp* polonies ● *gfp* polonies ● *gapdh* polonies ● *ACTB* polonies
● Red fluorescent signal ● Green fluorescent signal

No. of UEIs ■ No. of beacons ■ No. of edges ■ No. of targets

**Fig. 5 | Applying a minimum indirect path filter efficiently removes topological artifacts from experimental data. a–f,** Data loss is calculated as the total number of UEIs remaining for reconstruction after the applied filter. White scale bars: 100 μm. Striped scale bars: $1 L_{diff}$. **a,** Reference microscopy image. **b,** Reconstruction created from uncorrected and unfiltered data.

**c,e,** Reconstructions after read count filters of 3 (**c**) and 4 (**e**). **d,f,** Reconstructions after indirect path filters of 1 (**d**) and 2 (**f**). **g,** Indirect path distribution of length 3. **h,i,** Number of remaining UEIs, edges, and beacon and target polonies after the read count filter (**h**) and the indirect path filter (**i**). Figure adapted with permission from: **a–f,** adapted from ref. 7, Elsevier.

microscopy reaction, many of them had low read counts (Extended Data Fig. 10), and, as a result, only 70.6% ($6.39 \times 10^5/9.03 \times 10^5$) of all UEIs remained for reconstruction.

By contrast, applying a minimum indirect path cutoff of 1 greatly improved the reconstruction quality, while only removing 3.6% of all UEIs ($3.2 \times 10^4/9.03 \times 10^5$; Fig. 5d). The reconstruction quality further improved with an indirect path cutoff of 2 (removing 4.8% of all UEIs; Fig. 5f). Applying an even higher cutoff did not clearly further improve the reconstruction. Using an indirect path cutoff therefore removed disruptive edges more efficiently than a read count filter, allowing more data to be used for the resulting reconstruction.

We also attempted to split possible fused nodes in this dataset. The subgraphs formed from the nodes connected to any particular node often formed more than two connected components ($8.9 \times 10^4/1.1 \times 10^5$; 82.4%), similar to the simulated datasets with low amplitude and spread, meaning it could not be used for accurate partitioning. Indeed, when describing the experimental data in terms of amplitude and spread, we found it had low amplitude and average spread. The dimensions of the sample of an accurate reconstruction (Fig. 5f) were $\sim 9 \times 8 L_{diff}^2$ equivalents, that is, 11–13 times as long as the average distance of two connected nodes ($0.68 \pm 0.59 L_{diff}^2$ equivalents), suggesting an average to low spread. Of all $1.94 \times 10^7$ polony pairs that were within the average pairing distance, only $6.72 \times 10^5$ edges (3.5%) were obtained

from the sequencing data, similar to a low amplitude in the simulation. Possible fused nodes could therefore not be identified.

Overall, applying a minimum indirect path filter efficiently removed disruptive crosslinks from the experimental data, losing only a small percentage of all connections. Because the fidelity of the reconstruction scales with the amount of available data, the proposed algorithm could provide a useful filter to obtain more accurate reconstructions from sparse data.

## Discussion

We note a few observations, shortcomings and directions for future investigations. First, MinIPath did not exclusively or completely remove spurious crosslinks from simulated datasets, as the indirect path values for both partially overlapped, regardless of spread or amplitude. An adaptation of the algorithm taking longer indirect paths of five or seven steps may provide a better distinction between the erroneous and non-erroneous edges. The same principle could be applied to improve the performance of the node splitting algorithm on sparse data, to still allow one to find groups of indirectly connected nodes when fewer connections are present. However, such adaptations do come at an increased computational cost. Calculating the indirect paths of length $k$ has an expected runtime performance of $O(d^k \times |E|)$, where $d$ is the average degree, and $|E|$ is the number of edges in the connection graph.

Using longer indirect paths in the fused node algorithm will also result in more connections between the two groups of connected nodes, which might increase the number of false negatives. Such an adaptation would therefore have to be carefully characterized.

Second, it is possible that for edges connecting nodes in sparse areas of the samples (such as at the edge), lower indirect path values are calculated. To correct for these inaccuracies, the indirect path values would have to be corrected and normalized using the degree of each node. Such adaptations may be required for samples with a larger variation in the node density across the sample.

Finally, we have applied our error correction algorithm here on DNA microscopy data, but we note that the same principle could be applied on any dataset where adjacency is the primary source of data, such as Hi-C data[15]. Several methods have been described to obtain the 3D organization from pairing data between genomic regions[16], and it remains unclear, to our knowledge, how artifacts affect these. For this purpose, the method could be adapted to work on non-bipartite graphs. How errors affect these reconstructions and whether this correction algorithm can improve them remains a topic for future studies.

## Methods
### Algorithm description
As input, the algorithms take undirected, bipartite, weighted graphs, here called $G$:

$$G = (U, V, E, w) \tag{4}$$

$$E \subseteq U \times V \tag{5}$$

$$w(x, y) = \begin{cases} d \in \mathbb{N} & \text{if } (x, y) \in E \\ 0 & \text{if } (x, y) \notin E \end{cases} \tag{6}$$

$U$ and $V$ are the independent sets of nodes defining the bipartition of the graph, $E$ is the set of edges, and $w$ is a function assigning an edge weight $d$ to each connected pair of nodes ($d$ is not necessarily a constant, but varies per node pair).

Spurious crosslinks are identified by counting the number of short indirect paths between two nodes connected by an edge. We calculate the indirect path value at three steps ($w_{i3}$) by taking the product of the edge weights of each of the three edges that form one indirect path between the two nodes $x$ and $y$, then adding these products together for all indirect paths connecting the nodes:

$$w_{i3}(x, y) = \sum_{(a, b)|(a \in V; a \neq y; b \in U; b \neq x)} w(x, a) \times w(a, b) \times w(b, y) \tag{7}$$

For fused node correction, let $S_a$ denote the immediate neighbors of some node $a$. Although these nodes do not have edges among themselves (because they belong to the same bipartition), they can be indirectly connected at two steps. For convenience, we form the graph $G_a$ from the nodes $S_a$, the edges $E_a$ and the edge weights $w_{i2}$:

$$G_a = (S_a, E_a, w_{i2}) \tag{8}$$

where the edges $E_a \subseteq S_a^2$ are formed between the nodes in $S_a$, and the edge weights are given for each $(x, y) \in S_a^2$:

$$w_{i2}(x, y) = \begin{cases} \sum_{b \in U; b \neq a} w(b, x) \times w(b, y) & \text{if } a \in U \\ \sum_{b \in V; b \neq a} w(x, b) \times w(y, b) & \text{if } a \in V \end{cases} \tag{9}$$

Naturally, if $w_{i2}(x, y) = 0$, $x$ and $y$ do not have an edge. Note that, in contrast to $G$, $G_a$ is not a bipartite graph. Also, the edges to the original node $a$ are not used to find the edges in $w_{i2}$; that is, indirect rather than direct paths are used.

To obtain the partition of $G_a$, we first check whether $G_a$ is naturally disconnected into multiple components. If it consists of exactly two disconnected components, those are used as the partitions of $G_a$, with a normalized cut value of 0.0. If more than two components are found, the node is marked as unevaluable (due to sparse data). Otherwise, we apply spectral graph partitioning[13] to partition $G_a$ into two components, $S_{a1}$ and $S_{a2}$. The partition is evaluated by calculating the normalized cut[14]. The cut for this specific partition is first calculated by taking the sum of the edge weights removed between sets $S_{a1}$ and $S_{a2}$ by the partitioning:

$$\text{cut}(S_{a1}, S_{a2}) = \sum_{(u, v)|u \in S_{a1}; v \in S_{a2}} w_{i2}(u, v) \tag{10}$$

and the normalized cut is calculated by dividing the cut by the sum of the edge weights in each partition[14]:

$$\text{ncut}(S_{a1}, S_{a2}) = \frac{\text{cut}(S_{a1}, S_{a2})}{\sum_{(u, t)|u \in S_{a1}; t \in S_a} w_{i2}(u, t)} + \frac{\text{cut}(S_{a1}, S_{a2})}{\sum_{(u, t)|u \in S_{a2}; t \in S_a} w_{i2}(u, t)} \tag{11}$$

When the normalized cut is below the cutoff, node $a$ is removed, and two new nodes are created in $G$ that each inherit the edges of node $a$ to either the nodes in $S_{a1}$ or $S_{a2}$.

### Algorithm implementation
Two methods were implemented in Python to calculate weighted indirect paths. The first method starts from the asymmetric adjacency matrix $A(i, j) = w(i, j)$, then calculating the three-step adjacency matrix $A_3(i, j) = A \times A^T \times A$, then, for each node pair with an edge, subtracting the paths that use their direct edge, while setting other node pairs to 0:

$$A_{i3}(i, j) = \begin{cases} A_3(i, j) - w(i, j) \times \left( \sum_k w(i, k)^2 + \sum_k w(j, k)^2 - w_{ij} \right), & A(i, j) > 0 \\ 0, & A(i, j) = 0 \end{cases} \tag{12}$$

Calculating $A_3$ becomes computationally challenging for large datasets, partly because it calculates all paths of length three, not just those between nodes that were originally connected. We therefore implemented the calculation of only the indirect paths of length three between nodes connected in the original dataset, using sparse matrices. This method first calculates $A_2(i, j) = A \times A^T$, which contains all two-step paths between all nodes of one partition (for example, $U$). It then iterates over every edge to find all indirect paths of length three.

**Pseudocode**. 
```
For node a in V:
  For every node b connected to a (i.e., A(i_b, i_a) > 0):

    common nodes = set(nodes connected to b in two
    steps) U

               set(nodes connected to a)

               (that is, where(A_2(i_b, :) > 0) U
               where(A(:, i_a) > 0))

    For node c in common node indices:

      A_i3(i_c, i_a) += (A_2(i_b, i_c) - A(i_b, i_a) * A(i_c, i_a))
      * A(i_b, i_a)
```

Here $i_x$ denotes the index of node $x$. The second method is implemented in Python with Numba[17] acceleration to allow for the use of multiple threads.

For node splitting, the graph $G_a$ was extracted and partitioned using a spectral graph partitioning tool from the scikit-learn package[18]. Nodes were not considered for splitting if they had fewer than four connections, or if $G_S$ consisted of more than two components. Normalized cuts were calculated first for all nodes, then nodes were selected for splitting according to the applied cutoff. If two nodes with an edge were both split, the edge was removed.

## Simulated data processing

Polony locations were reconstructed using the sMLE method described previously[7]. A slightly adapted version of the pipeline was used to process large amounts of files more easily. In contrast to experimental data, simulated data were not subjected to the iterative minimum UEI filter before reconstruction. Analysis was done with custom scripts in Python v3.9.12, using the packages numba v.0.53.1 (ref. 17), numpy v1.22.4 (ref. 19), pandas v.1.4.4 (ref. 20), scikit-learn v1.1.3 (ref. 18) and scipy v1.9.3 (ref. 21), and visualized with seaborn v.0.11.2 (ref. 22). Reconstructions from graphs where the largest connected component was smaller than 80% of all nodes were not considered for further analysis. Global reconstruction quality was assessed using the Procrustes disparity, and local reconstruction quality was assessed by the overlap of the $k$-nearest neighbors for each node in the original and reconstruction positions, as suggested previously[9].

To evaluate the node splitting accuracy, we paired each set of nodes $S_{a1}$ and $S_{a2}$ to their closest match among the two sets of nodes in $S_b$ and $S_c$ (i.e. the nodes connected to the original nodes $b$ and $c$ that made up the fused node), and calculated the overlap:

$$\text{Overlap} = \frac{\sum_{n \in \{1,2\}} \max\left(\frac{S_{an} \cap S_b}{S_{an}}, \frac{S_{an} \cap S_c}{S_{an}}\right)}{2} \tag{13}$$

## Experimental data processing

Raw sequencing data were downloaded from the Sequencing Read Archive (project no. PRJNA487001, sample 3), and processed as described previously, without a minimum read count. After then applying either a read count filter or an indirect path filter, the remaining nodes were filtered as described earlier[7], first by iteratively removing nodes with fewer than two associated products to remove possible uncorrected sequencing errors, then by selecting the largest connected component. Properties displayed in Fig. 5h,i are derived from the graphs after all filters were applied.

## Reporting summary

Further information on research design is available in the Nature Portfolio Reporting Summary linked to this Article.

## Data availability

The input for the simulations and the data generated by the simulations are available in the Zenodo repository at https://doi.org/10.5281/zenodo.10256692 ref. 23. The previously published raw experimental data[7] are available at the Sequencing Read Archive (project no. PRJNA487001, sample 3). Source data are provided with this paper.

## Code availability

The code used for the simulation, error correction and imaging, as well as the adapted reconstruction code, can all be found in the Zenodo repository (https://doi.org/10.5281/zenodo.10256692 ref. 23). The code for error correction is also available on GitHub at https://github.com/Alexamk/minipath/.

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

## Acknowledgements

We acknowledge support from the Knut and Alice Wallenberg Foundation for B.H. (grants nos. KAW 2017.0114 and KAW 2017.0276), from the European Research Council (ERC) for B.H. (acronyms: Cell

Track, GA no. 724872 and qScope, GA no. 101097367), from the Swedish Research Council for B.H. (grant no. 2019-01474), from the Göran Gustafsson foundation for B.H., and from the European Commission H2020 MSCA ITN (DNA Robotics GA no. 765703) for B.H. and I.B.

## Author contributions

Conceptualization was provided by A.M.K. and B.H., methodology by A.M.K., investigations by A.M.K. and I.B., formal analysis by A.M.K., and visualization by A.M.K. The original draft was written by A.M.K, and review and editing by A.M.K., I.B. and B.H. Funding acquisition was performed by B.H.

## Funding

## Competing interests

The authors declare no competing interests.

## Additional information

**Extended data** is available for this paper at https://doi.org/10.1038/s43588-023-00589-x.

**Correspondence and requests for materials** should be addressed to Björn Högberg.

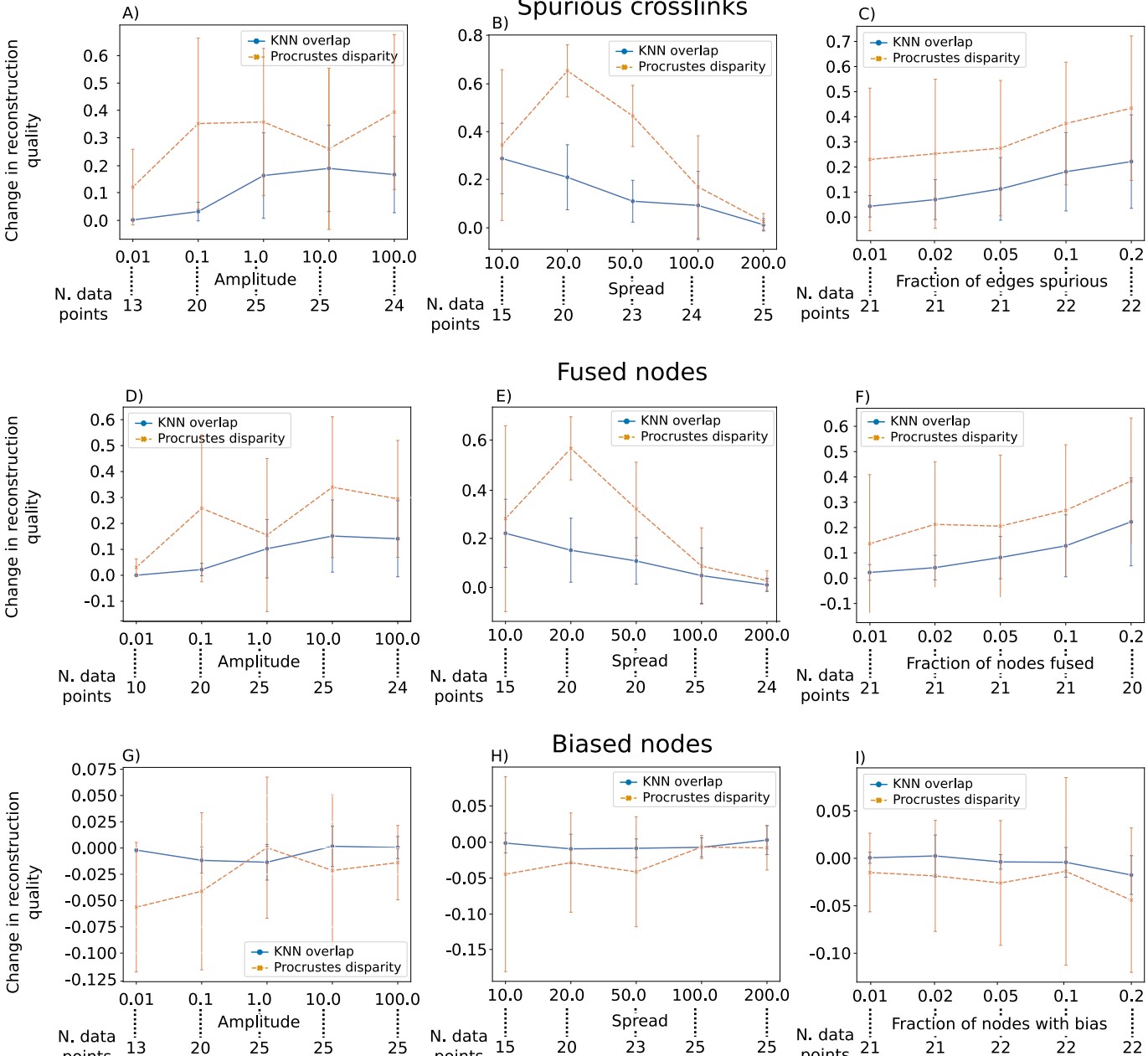

**Extended Data Fig. 1 | The robustness of reconstructions to introduced errors depends on the amplitude, spread and number of errors.** Mean reconstruction quality when adding spurious crosslinks (A-C), fusing randomly selected nodes (D-F), or applying a 2x amplitude bias to randomly selected nodes (G-I). Error bars shown represent the standard deviation. Number of samples per mean given below each figure.

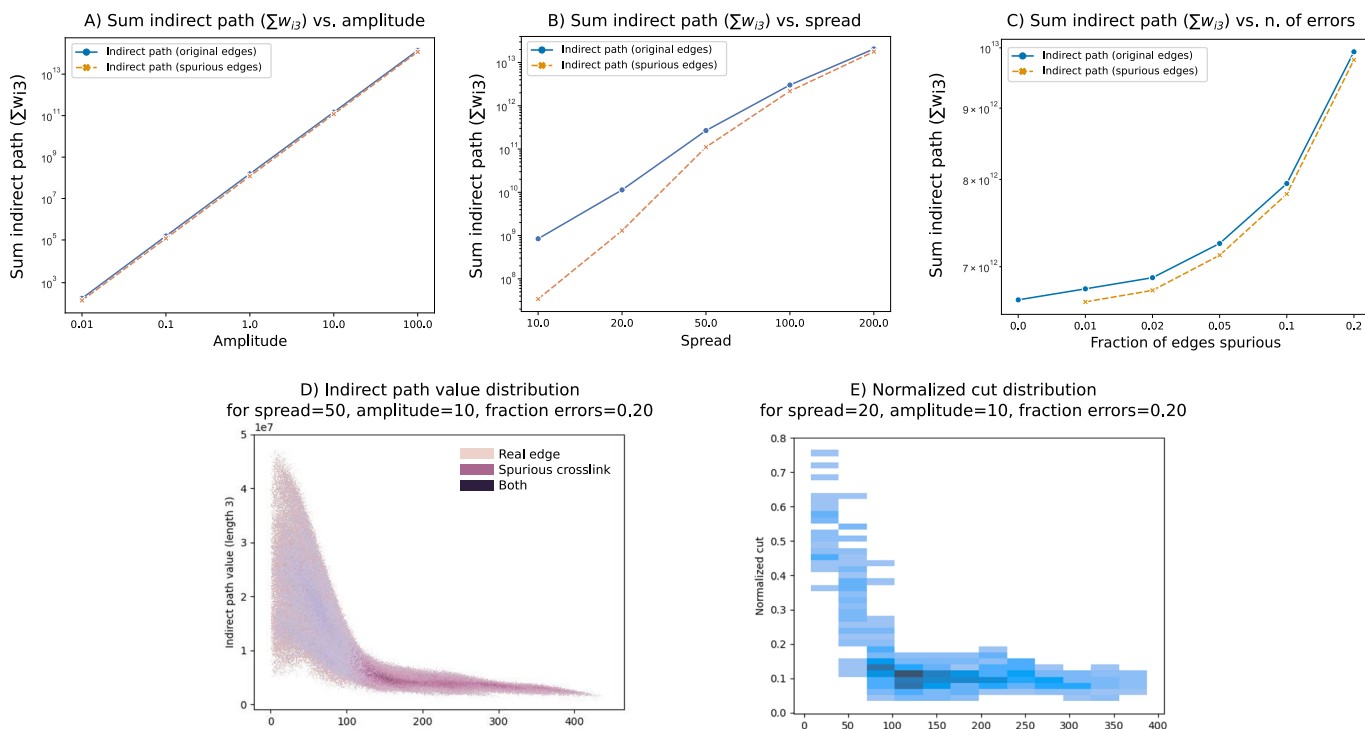

**Extended Data Fig. 2 | The indirect path value of length 3 depends on amplitude, spread, number of errors and distance.** Dependence of indirect path value on (A) amplitude, (B) spread, and (C) number of errors.

(D) Dependence of indirect path value on distance shown in a single simulation (amplitude of 10, spread of 50, 20% errors). E) Dependence of normalized cuts shown in a single simulation (amplitude of 10, spread of 20, 20% errors)

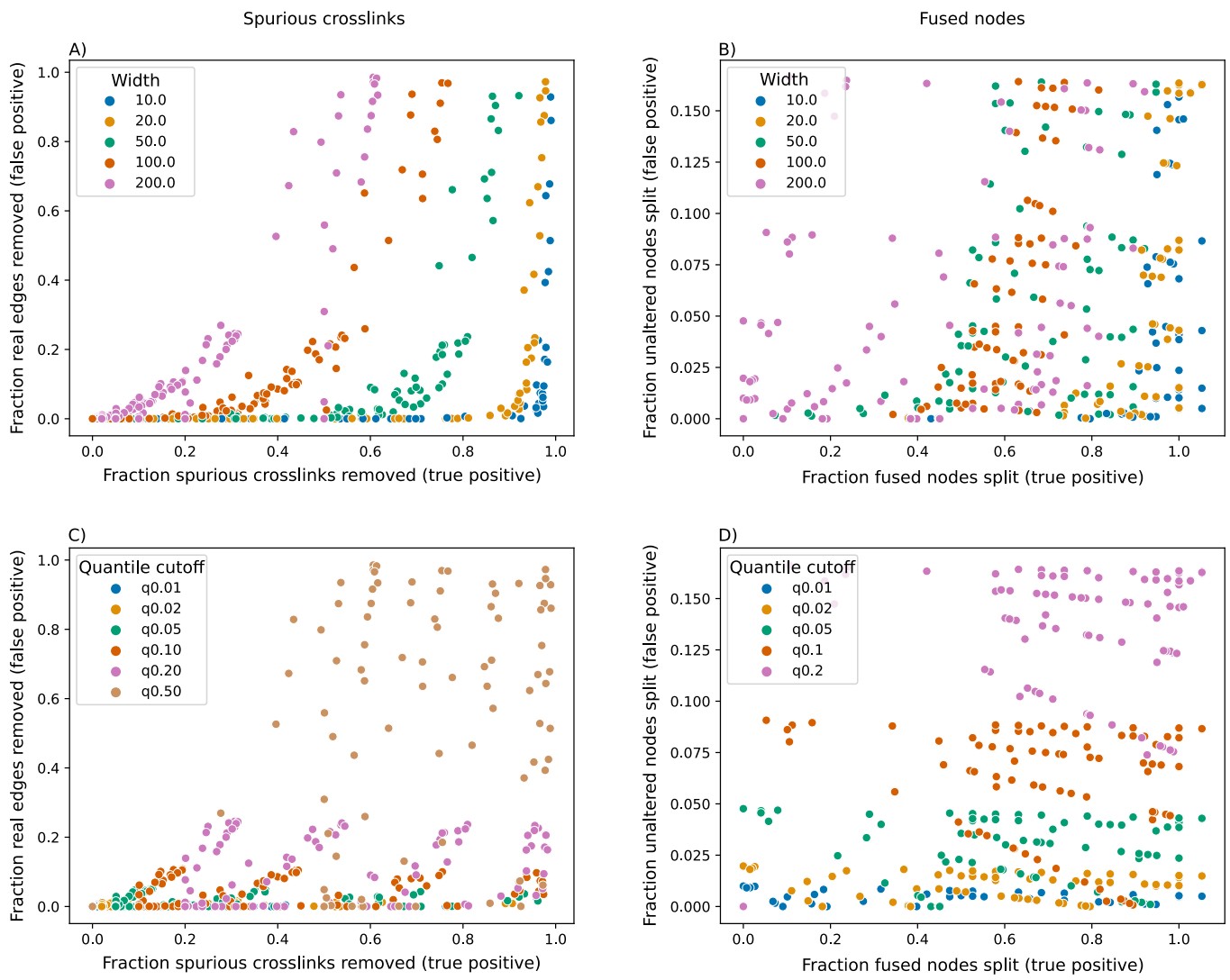

**Extended Data Fig. 3 | True positive versus false positive rate primarily depends on spread.** Each point in each scatterplot represents the true positive and false positive rate in a single simulation where errors were added and corrected. (A, B) are colored by spread, (C, D) are colored by cutoff.

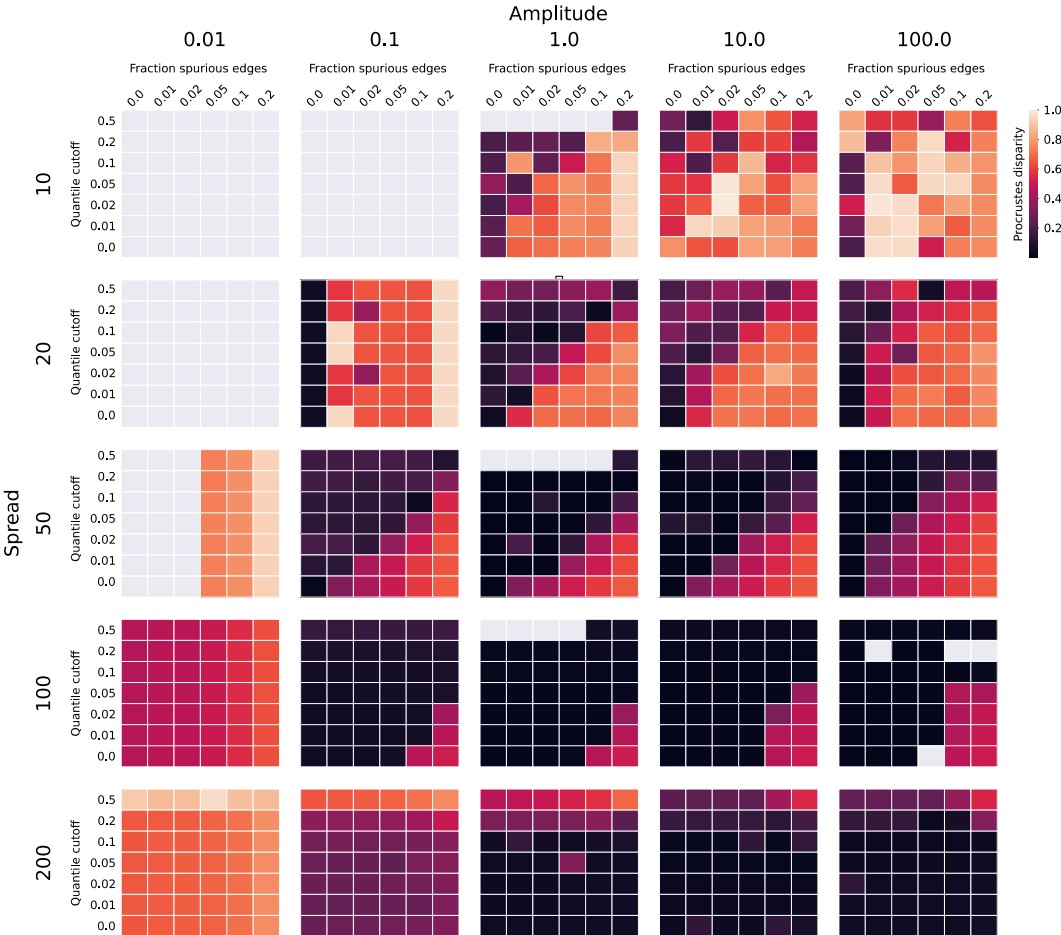

**Extended Data Fig. 4 | Procrustes disparity of all simulations containing spurious crosslinks.** Procrustes disparity is shown after different amounts of spurious crosslinks are added and corrected.

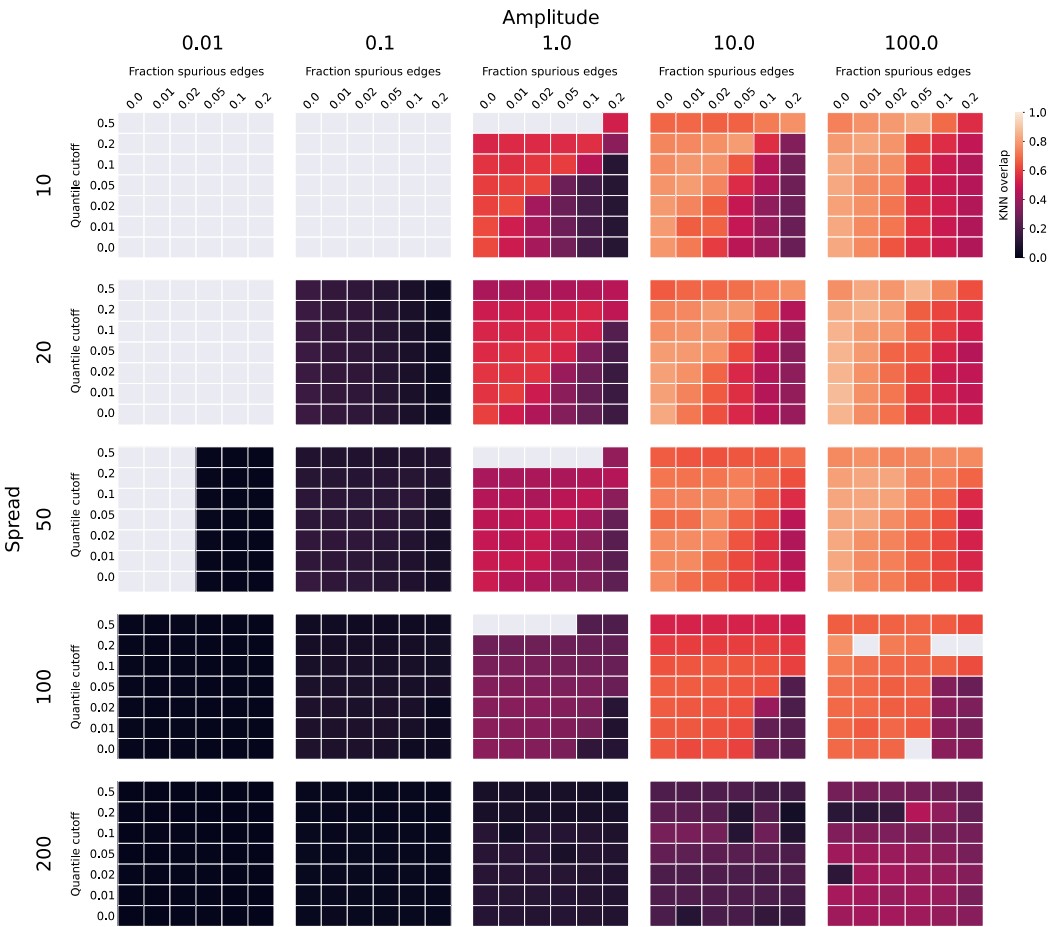

**Extended Data Fig. 5 | KNN overlap of all simulations containing spurious crosslinks.** KNN overlap value is shown after different amounts of spurious crosslinks are added and corrected.

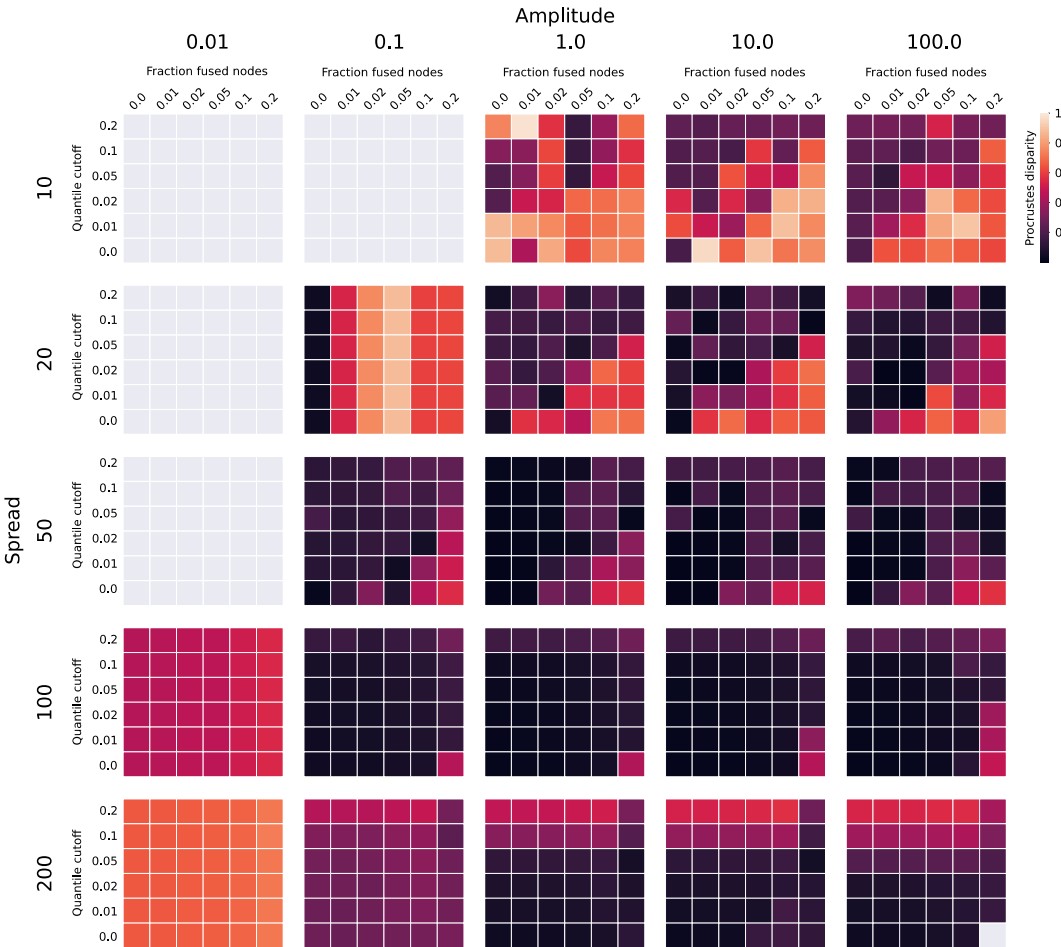

**Extended Data Fig. 6 | Procrustes disparity of all simulations containing fused nodes.** Procrustes disparity is shown after different amounts of fused nodes are added and corrected.

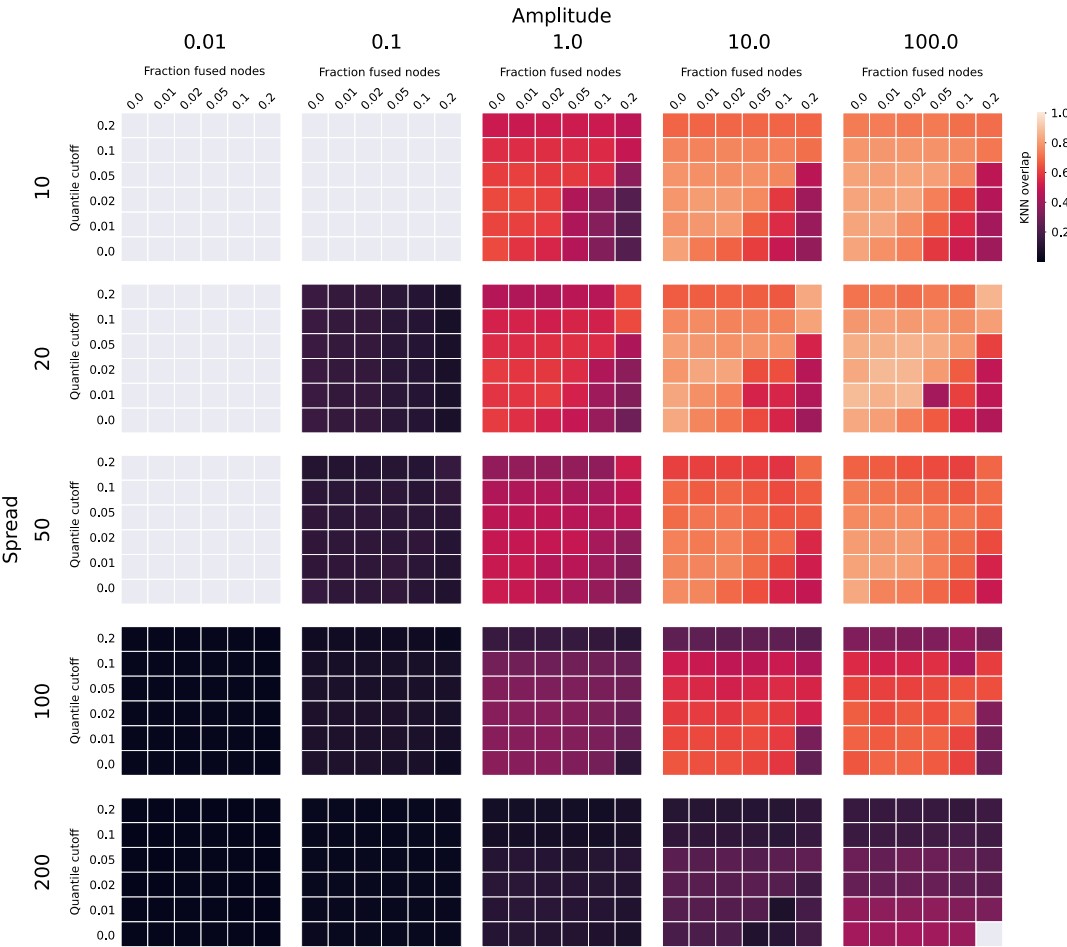

**Extended Data Fig. 7 | KNN overlap of all simulations containing fused nodes.** KNN overlap is shown after different amounts of fused nodes are added and corrected.

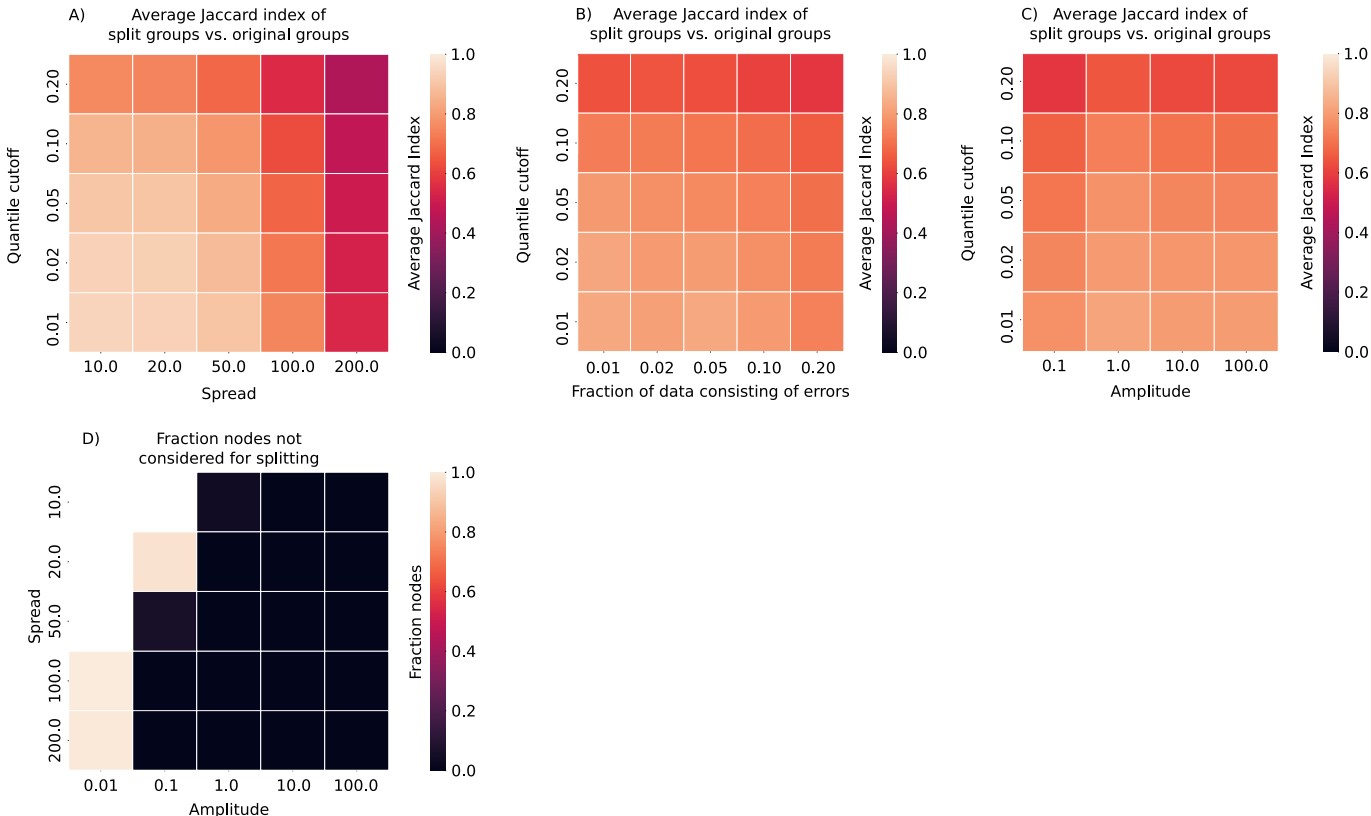

**Extended Data Fig. 8 | Further details on node splitting accuracy.** (A-C) Heatmaps show the mean node splitting accuracy as dependent on (A) spread, (B) fraction of errors or (C) the amplitude. (D) Fraction of unfused nodes not considered for splitting in data without errors.

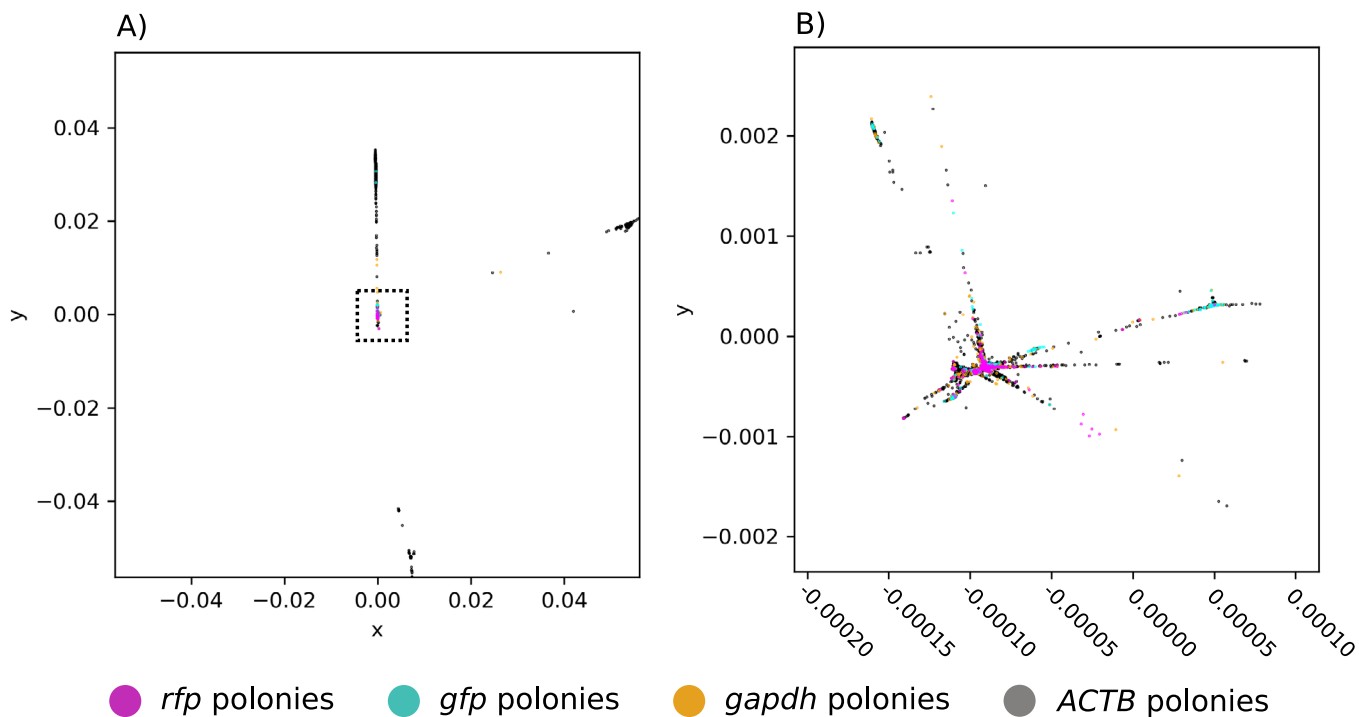

**Extended Data Fig. 9 | Not correcting experimental datasets results in collapsed reconstructions.** X and y scale represent the x and y coordinates of the polonies after sMLE reconstruction, respectively. A) Full scale. B) Zoomed in.

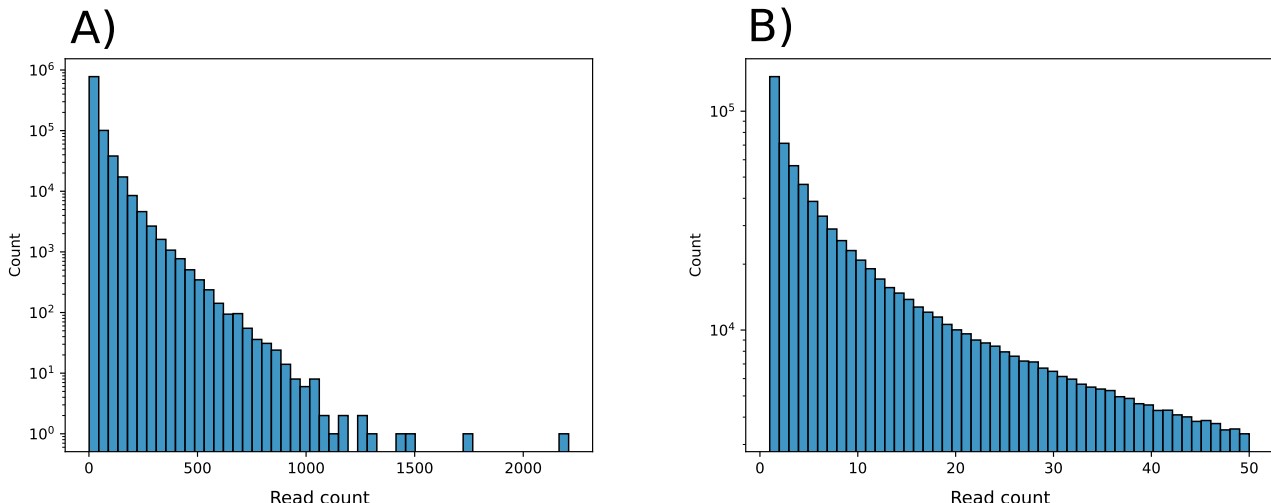

**Extended Data Fig. 10 | Read count distribution of the analyzed experimental dataset.** Read counts of all the unique products connecting two polonies in the experimental dataset. A) Full scale. B) Zoomed in on the range 1-50.

# Reporting Summary

## Statistics

For all statistical analyses, confirm that the following items are present in the figure legend, table legend, main text, or Methods section.

| n/a | Confirmed | |
|---|---|---|
| ☐ | ☒ | The exact sample size (*n*) for each experimental group/condition, given as a discrete number and unit of measurement |
| ☒ | ☐ | A statement on whether measurements were taken from distinct samples or whether the same sample was measured repeatedly |
| ☒ | ☐ | The statistical test(s) used AND whether they are one- or two-sided *Only common tests should be described solely by name; describe more complex techniques in the Methods section.* |
| ☒ | ☐ | A description of all covariates tested |
| ☒ | ☐ | A description of any assumptions or corrections, such as tests of normality and adjustment for multiple comparisons |
| ☐ | ☒ | A full description of the statistical parameters including central tendency (e.g. means) or other basic estimates (e.g. regression coefficient) AND variation (e.g. standard deviation) or associated estimates of uncertainty (e.g. confidence intervals) |
| ☒ | ☐ | For null hypothesis testing, the test statistic (e.g. *F*, *t*, *r*) with confidence intervals, effect sizes, degrees of freedom and *P* value noted *Give P values as exact values whenever suitable.* |
| ☒ | ☐ | For Bayesian analysis, information on the choice of priors and Markov chain Monte Carlo settings |
| ☒ | ☐ | For hierarchical and complex designs, identification of the appropriate level for tests and full reporting of outcomes |
| ☒ | ☐ | Estimates of effect sizes (e.g. Cohen's *d*, Pearson's *r*), indicating how they were calculated |

*Our web collection on statistics for biologists contains articles on many of the points above.*

## Software and code

Policy information about availability of computer code

| | |
|---|---|
| Data collection | Data was generated using only custom code provided with the manuscript, and is made available at 10.5281/zenodo.10256692. Simulation code was ran in python v3.9.12 using numba v.0.53.1 and numpy v1.22.4. |
| Data analysis | Data was analysed using custom code provided with the manuscript, availble from 10.5281/zenodo.10256692 and also from https://github.com/Alexamk/minipath/). Analysis code was ran in python v3.9.12 using numba v.0.53.1, numpy v1.22.4, pandas v.1.4.4, scikit-learn v1.1.3, scipy v1.9.3, and seaborn v.0.11.2.<br><br>Part of the code used to create the microscopy reconstructions is slightly adapted from previously published code, available from https://github.com/jaweinst/dnamic. |

For manuscripts utilizing custom algorithms or software that are central to the research but not yet described in published literature, software must be made available to editors and reviewers. We strongly encourage code deposition in a community repository (e.g. GitHub). See the Nature Portfolio guidelines for submitting code & software for further information.

## Data

Policy information about [availability of data](availability of data)

All manuscripts must include a [data availability statement](data availability statement). This statement should provide the following information, where applicable:
- Accession codes, unique identifiers, or web links for publicly available datasets
- A description of any restrictions on data availability
- For clinical datasets or third party data, please ensure that the statement adheres to our [policy](policy)

> The input for the simulations and the data generated by the simulations are available in the Zenodo repository: 10.5281/zenodo.10256692. The previously published raw experimental data is available at the Sequencing Read Archive (project number PRJNA487001, sample 3). Source Data for Figures 2, 4, and 5, and for Extended Data Figure 1-10 is available with this manuscript.

## Human research participants

Policy information about [studies involving human research participants and Sex and Gender in Research.](studies involving human research participants and Sex and Gender in Research.)

| | |
|---|---|
| Reporting on sex and gender | n/a |
| Population characteristics | n/a |
| Recruitment | n/a |
| Ethics oversight | n/a |

Note that full information on the approval of the study protocol must also be provided in the manuscript.

# Field-specific reporting

Please select the one below that is the best fit for your research. If you are not sure, read the appropriate sections before making your selection.

☒ Life sciences ☐ Behavioural & social sciences ☐ Ecological, evolutionary & environmental sciences

For a reference copy of the document with all sections, see [nature.com/documents/nr-reporting-summary-flat.pdf](nature.com/documents/nr-reporting-summary-flat.pdf)

# Life sciences study design

All studies must disclose on these points even when the disclosure is negative.

| | |
|---|---|
| Sample size | Simulations were run once per condition, since we expected variation when changing simulation parameters not to affect the overall results. We instead fixed one simulation parameter at a time and varied the others to obtain multiple samples per parameter, from which means were calculated.<br>No experimental data was collected for which choosing a sample size may have been appropriate. |
| Data exclusions | Reconstructions were left out of the data analysis if the largest connected component of the corresponding neighborhood graph contained less than 80% of all nodes, since the reconstruction accuracy could not be accurately determined for these reconstructions |
| Replication | Simulations were not repeated, since we expected variation when changing simulation parameters not to affect the overall results.<br>We instead fixed one simulation parameter at a time and varied the others to obtain multiple samples per parameter.<br>No experimental data was collected for which replication may have been appropriate. |
| Randomization | No experiment was performed for which randomization was applicable. |
| Blinding | No experiment was performed for which blinding was applicable. |

# Reporting for specific materials, systems and methods

We require information from authors about some types of materials, experimental systems and methods used in many studies. Here, indicate whether each material, system or method listed is relevant to your study. If you are not sure if a list item applies to your research, read the appropriate section before selecting a response.

## Materials & experimental systems

| n/a | Involved in the study |
|---|---|
| ☒ | ☐ Antibodies |
| ☒ | ☐ Eukaryotic cell lines |
| ☒ | ☐ Palaeontology and archaeology |
| ☒ | ☐ Animals and other organisms |
| ☒ | ☐ Clinical data |
| ☒ | ☐ Dual use research of concern |

## Methods

| n/a | Involved in the study |
|---|---|
| ☒ | ☐ ChIP-seq |
| ☒ | ☐ Flow cytometry |
| ☒ | ☐ MRI-based neuroimaging |

