## [Peer Review File · Nature Computational Science]

Peer Review Information

Journal: Nature Computational Science

Manuscript Title: An error correction strategy for image reconstruction by DNA sequencing microscopy

Corresponding author name(s): Professor Björn Högberg

Editorial Notes:

Reviewer Comments & Decisions:

Decision Letter, initial version:

Date: 6th October 23 12:18:53
Last Sent: 6th October 23 12:18:53
Triggered By: Ananya Rastogi
From: ananya.rastogi@nature.com
To: bjorn.hogberg@ki.se
BCC: ananya.rastogi@nature.com
Subject: Decision on Nature Computational Science manuscript NATCOMPUTSCI-23-0819A
Message: ** Please ensure you delete the link to your author homepage in this e-mail if you wish to forward it to your co-authors. **

Dear Professor Högberg,

Your manuscript "An error correction strategy for image reconstruction by DNA sequencing microscopy" has now been seen by 3 referees, whose comments are appended below. You will see that while they find your work of interest, they have raised points that need to be addressed before we can make a decision on publication.

The referees' reports seem to be quite clear. Naturally, we will need you to address *all* of the points raised.

While we ask you to address all of the points raised, the following points need to be substantially worked on:

- Please explore if the eigenvector calculation might improve with better normalization.
- Please rewrite the manuscript to simplify the information on the 2D grid model and

the derivation through graph theory.

Please use the following link to submit your revised manuscript and a point-by-point response to the referees' comments (which should be in a separate document to any cover letter):

[REDACTED]

** This url links to your confidential homepage and associated information about manuscripts you may have submitted or be reviewing for us. If you wish to forward this e-mail to co-authors, please delete this link to your homepage first. **

To aid in the review process, we would appreciate it if you could also provide a copy of your manuscript files that indicates your revisions by making use of Track Changes or similar mark-up tools. Please also ensure that all correspondence is marked with your Nature Computational Science reference number in the subject line.

In addition, please make sure to upload a Word Document or LaTeX version of your text, to assist us in the editorial stage.

To improve transparency in authorship, we request that all authors identified as 'corresponding author' on published papers create and link their Open Researcher and Contributor Identifier (ORCID) with their account on the Manuscript Tracking System (MTS), prior to acceptance. ORCID helps the scientific community achieve unambiguous attribution of all scholarly contributions. You can create and link your ORCID from the home page of the MTS by clicking on 'Modify my Springer Nature account'. For more information please visit www.springernature.com/orcid.

We hope to receive your revised paper within three weeks. If you cannot send it within this time, please let us know.

Best regards,

Ananya Rastogi, PhD
Senior Editor
Nature Computational Science

Reviewers comments:

Reviewer #1 (Remarks to the Author):

This is a thoughtful paper about a type of noise prevalent in DNA microscopy data sets. It is a meaningful contribution to the so-far thin literature on this growing subject, and so I believe it merits publication. That being said, there are a couple issues that it would be helpful for the authors to address.

1. What seems to be a missed opportunity is dealing with the effect of a long-tailed distribution of edges-per-node in the fused-node case. Namely, we expect there to be a few nodes with a high degree of connectivity versus most that have a low degree of connectivity. Because the graph-cutting algorithm the authors have devised appears to only normalize edge-weights connected to the (putatively) fused node "a", it seems miss the opportunity of weeding out edges that belong to over-connected nodes (those at the "tail" of the degree distribution) for which most edges lie outside of the locally-defined sub-graph around "a".

2. The authors compare filtering on indirect-connections versus read-counts and show an improvement in data retention. This is useful, but does potentially point to a work-around that it would be good to at least get some discussion on (if not some additional analysis). Ostensibly, constraining the inferred position solution to the top eigenvectors of the UEI/adjacency matrix would implicitly maximize the prominence of the highly connected portions of the data set. To the extent that the more "surgical" approach the authors take is better, one might wonder if the eigenvector calculation itself might improve with better normalization (and avoid the current apparent drawback of the authors' method requiring calculation on a much-less-sparse matrix than what is present in the raw data set). For example, as pointed out above, the long-tailed distribution of degrees-per-node might be having an undue influence on these data sets. De-weighting edges that connect to nodes of high non-local connectivity may have a similar effect as the authors' procedure (and would be computationally more efficient).

Reviewer #1 (Remarks on code availability):

I have reviewed the code but have not independently run it.

Reviewer #2 (Remarks to the Author):

The authors consider a novel DNA sequencing -based nonoptical microscopy technique, whereby individually barcoded short DNA strands are first randomly deposited onto a primer surface and then locally amplified to form labeled polymer colonies or "colonies". Crosslinked concatemers from nearby colonies then provide information about adjacencies of the seed-strand primer sites which can be used, post-sequencing, to reconstruct the spatial arrangement of the seed locations on the primer lawn.

The reconstruction can however be distorted by at least two types of experimental artefacts: (i) spurious crosslinks fabricated by incomplete PCR in the postprocessing phase and (ii) colonies which are fused together because of incidentally identical seed-strand barcodes, due to either coincidence or as a by-product of sequencing-error correction.

The manuscript presents computational methods for identifying and pruning these artefacts by: (i) validating all crosslinks by checking for the presence of matching short indirect connecting paths in the colony interconnection network and (ii) checking for the cohesion of colony neighborhoods by determining minimal-cost partitions for them. (The premise being that the network neighborhoods of artificially fused colonies would easily partition into components, whereas the network neighborhoods of true colonies

would be densely connected.)

The methods are validated on a few numerical datasets and on one experimental dataset from an earlier publication. The numerical results look quite promising, and on the experimental dataset an error-correction quality comparable to earlier results is achieved with significantly reduced data loss, i.e. smaller number of pruned connections. (The captioning of the figures reporting the results could still be improved, though, particularly in the Extended Data section where it is not always clear which data sets are being considered and how the figures are to be interpreted.)

In general, I thus think that the manuscript addresses an essential concern in the development of a highly promising new technology in a tentatively promising manner.

However I was quite confused about the way the material is presented. The first Section of the "Results" chapter of the manuscript outlines the proposed error-correction methods in the toy model of a complete bipartite 2D grid graph, which is neither a good proxy for the actual polony interconnection networks, nor even close to the actual graph model used in the authors' own implementation of their methods, as discussed in the second Section of the "Results" chapter. (These issues are discussed more extensively in my attached "Detailed comments".)

The relationship of these two Sections is not explained to the reader, and in retrospect I actually find the whole presentation in terms of the 2D grid model superfluous and misleading. (Moreover this first Section contains a number of mathematical inconsistencies and inaccuracies which at points make following the argumentation there quite difficult; cf. again my attached "Detailed comments".)

I would thus propose that the authors carry out a fundamental rewrite of the manuscript, where they eliminate the confusing and misleading diversion via the 2D grid model and present their methods directly in terms of their actual diffusion-parameter based weighted-graph model. Such a rewrite would greatly improve the structure and readability of the manuscript, and as far as I can see is completely doable even following the general storyline of the present first Section but working in the framework of the actual weighted-graph model rather than the grid graph toy model.

Review report on NATCOMPUTSCI-23-0819A, “An error correction strategy for image reconstruction by DNA sequencing microscopy”

The authors consider a novel DNA sequencing -based nonoptical microscopy technique, whereby individually barcoded short DNA strands are first randomly deposited onto a primer surface and then locally amplified to form labeled polymer colonies or “polonies”. Crosslinked concatemers from nearby polonies then provide information about adjacencies of the seed-strand primer sites which can be used, post-sequencing, to reconstruct the spatial arrangement of the seed locations on the primer lawn.

The reconstruction can however be distorted by at least two types of experimental artefacts: (i) spurious crosslinks fabricated by incomplete PCR in the postprocessing phase and (ii) polonies which are fused together because of incidentally identical seed-strand barcodes, due to either coincidence or as a by-product of sequencing-error correction.

The manuscript presents computational methods for identifying and pruning these artefacts by: (i) validating all crosslinks by checking for the presence of matching short indirect connecting paths in the polony interconnection network and (ii) checking for the cohesion of polony neighborhoods by determining minimal-cost partitions for them. (The premise being that the network neighborhoods of artificially fused polonies would easily partition into components, whereas the network neighborhoods of true polonies would be densely connected.)

The methods are validated on a few numerical datasets and on one experimental dataset from an earlier publication. The numerical results look quite promising, and on the experimental dataset an error-correction quality comparable to earlier results is achieved with significantly reduced data loss, i.e. smaller number of pruned connections. (The captioning of the figures reporting the results could still be improved, though, particularly in the Extended Data section where it is not always clear which data sets are being considered and how the figures are to be interpreted.)

In general, I thus think that the manuscript addresses an essential concern in the development of a highly promising new technology in a tentatively promising manner.

However I was quite confused about the way the material is presented. The first Section of the “Results” chapter of the manuscript outlines the proposed error-correction methods in the toy model of a complete bipartite 2D grid graph, which is neither a good proxy for the actual polony interconnection networks, nor even close to the actual graph model used in the authors’ own implementation of their methods, as discussed in the second Section of the “Results” chapter. (These issues are discussed more extensively in my “Detailed comments” below.)

The relationship of these two Sections is not explained to the reader, and in retrospect I actually find the whole presentation in terms of the 2D grid model superfluous and misleading. (Moreover this first Section contains a number of mathematical

inconsistencies and inaccuracies which at points make following the argumentation there quite difficult; cf. again my “Detailed comments” below.)

I would thus propose that the authors carry out a fundamental rewrite of the manuscript, where they eliminate the confusing and misleading diversion via the 2D grid model and present their methods directly in terms of their actual diffusion-parameter based weighted-graph model. Such a rewrite would greatly improve the structure and readability of the manuscript, and as far as I can see is completely doable even following the general storyline of the present first Section but working in the framework of the actual weighted-graph model rather than the grid graph toy model.

Detailed comments:

Line 20: 'improvement' → 'improvements'

Line 23: 'organization' → 'organization of'

Line 30: 'molecules use that' → 'molecules and use that'. (Or even better 'and use the pairing information'.)

Line 32: 'in tissue of interest' → 'in the tissue of interest'?

Line 38: 'it' → 'the method'

Line 50: 'misstakenly' → 'mistakenly'

Line 80: 'their' → 'its'

Line 81: 'extend' → 'extent'

Line 83: More simply $E \subseteq U \times V$

Line 84: What is the value of the weight parameter d here? From line 80 one would conclude that it is a constant $d = 1$ and the graph G is thus essentially unweighted. Also the rest of the section seems to support this interpretation. (Cf. e.g. Figure 2.)

Line 86: 'mapping function' → 'function'

Line 96: 'mapping function' → 'function'

Line 95: 'neighbors' or 'neighbours'? (Both spellings are now being used, but in general the manuscript seems to favor the American variety.)

Line 95: The expression 'indirect path within three steps' seems a bit unconventional. A more common version would be 'indirect path of length 3'.

Lines 99–100: 'indirect paths connecting each edge in three steps' → 'for each edge the number of indirect paths of length 3 connecting its endpoints'

Line 101: The equation (definition) on this line has no lead text, so comes up quite abruptly. Possibly the sentence on line 100 should end in a colon rather than a period? Also it would be somewhat more natural to write the product term in the summation as $w(x,a) * w(a,b) * w(b,y)$ rather than $w(x,a) * w(b,a) * w(b,y)$.

Lines 102–105: These properties of the pathcount weights $w_{i3}(x,y)$ are of course guaranteed only in the case of the perfect bicolored 2D grid graphs presented in Figure 2A. In other types of graphs, even ones which would be quite natural as models of polony networks, these properties may or may not hold to varying degrees. Hence any conclusions drawn from these characteristics need to be taken with a grain of salt.

Lines 107–108: So in graph-theoretic terms the set S_a comprises the (immediate) neighbors of node a ?

Line 111: What does the index S in the notation G_S stand for? Since this is the neighborhood graph of node a , notation G_a would seem more natural. Maybe also E_a rather than E_S ?

Line 113: The condition " $(x,y) \subseteq S_a^2$ " should be " $(x,y) \in S_a^2$ ". (Even more naturally, the equation could have as lead text "where for each $(x,y) \in S_a^2$ " and then as l.h.s. just ' $w_{i2}(x,y) = \dots$ ')"

Line 114: Since either $S_a \subseteq U$ or $S_a \subseteq V$, it is the case that $E_S = \emptyset$. So what is the intended correct definition of the edge set E_S ?

Line 116: According to the given definitions, the graph G_S is always totally disconnected, i.e. $E_S = \emptyset$. Please provide the correct definitions.

Line 133: This definition only gives the cut value for a specific partition $S_a = S_{a1} \cup S_{a2}$. Don't you eventually need something like:

$$\text{cut}(S_a) = \min_{(S_{a1}, S_{a2})} \{\text{cut}(S_{a1}, S_{a2}) | \dots\}?$$

Line 136: Same question as above for the $ncut$ measure. Also a general question: can a fused node have more than 2 constituents? How does the $ncut$ measure behave then?

Line 167: 'location' → 'locations'

- Line 171:** '2D grid of nodes' → '2D grid of nodes of two types'?
'randomly distributed a shape' → 'within a shape'/'according to a shape'?
- Lines 172–173:** 'the edge weight of any two nodes' → 'the weight of a connection/an edge between any two nodes'
- Line 174:** The connection model here seems to be quite different from what was discussed in the previous section: the network appears now to be fully connected, but with Poisson-distributed weights where the Poisson parameter depends on the characteristics of the diffusion model targeted in the numerical experiments. This is quite confusing: if the edge weights are not unit-valued, then the notion of 'indirect 3-step paths' discussed earlier becomes difficult to interpret. Also the structure of the network is quite different from what was discussed previously, and hence many of the arguments used to derive the error-correction methods there become moot. (The methods may still of course work well as heuristics, albeit in a less-justified way.) Please clarify the situation here.
- Line 183:** Still related to the previous comment: in the earlier discussion it was assumed that the polony graph is strictly bipartite. How is bipartiteness defined with respect to the present connection model? E.g. as far as I can see, with Poisson-distributed weights the resulting graph can contain odd-length cycles, which contradicts the bipartiteness condition. An obvious way around this problem would be to have two types of seed nodes, but this approach is not mentioned in the present discussion.
- Lines 210–212:** This graded method for pruning spurious crosslinks differs from the one presented earlier, where a threshold value of zero indirect paths of length 3 between nodes was used to identify spurious connections. Using a graded method is of course fine (and apparently necessary) for the present more general network model; however introducing this modification in such an offhanded way raises questions about (a) the relevance of the earlier discussion and (b) the correct choice of the threshold parameter in this new setting, where the network connectivity depends on the characteristics of the diffusion model.
- Line 219:** Again, how to choose the proper threshold value for the *ncut* measure in the present network model? How does this depend on the Poisson connectivity parameter or the underlying diffusion parameters?
- Line 296–306:** This paragraph seems to suggest that the *ncut* measure could not be applied to identify fused nodes for the available experimental dataset. Please indicate what improvements would be required to make the method applicable to real, in addition to simulated data?

Lines 338–339: This discussion seems to overestimate the performance challenges of testing for longer connecting paths. The neighborhood graphs are (for a constant Poisson parameters at least) typically very sparse, with some constant average degree d : e.g. in the grid graph of Figure 2A, $d = 4$. Then the expected runtime for enumerating all connecting paths of at most a fixed length k is just $O(d^k \times E) = O(E)$. (In fact for the grid graph of Figure 2A, the number of paths of length at most k starting from any node is just $O(k^2)$, and I suppose the same is true for any plane graph with unit-length edges.)

Reviewer #3 (Remarks to the Author):

Dear Authors,

I was pleasantly surprised that withing such a hot topic as DNA-microscopy such insightful additions could be made, and read your manuscript with great interest. Overall, I am convinced that the proposed method is a highly valuable addition to the technique and will serve to broaden the applicability of the method in the face of less-than-perfect experimental conditions.

The only major remark I have is that the novelty and impact of the method are to some extent lost because of the unnecessarily complex introduction and derivation trough graph theory.

Therefore it is my recommendation that lines 76-157 are rewored, and moved to the appendix/SI, and in their place a short paragraph is put clearly outlining how the procedure works, with frequent reference to figure 2 and providing an intuitive understanding of the 2 problems and why they can be detected with the proposed metrics.

Furthermore, as a minor remark I would like a further explanation on the treatment of edge nodes (at the border of the cell) and if the method could be altered to stop potential artifacts at these nodes which naturally have fewer connections

Reviewer #3 (Remarks on code availability):

It appears as though the code was all present, and a suitable readme file was included. However, due to time constrains I did not install the required packages or run the code

Author Rebuttal to Initial comments

Reviewers comments:

Reviewer #1 (Remarks to the Author):

This is a thoughtful paper about a type of noise prevalent in DNA microscopy data sets. It is a meaningful contribution to the so-far thin literature on this growing subject, and so I believe it merits publication. That being said, there a couple issues that it would be helpful for the authors to address.

1. What seems to be a missed opportunity is dealing with the effect of a long-tailed distribution of edges-per-node in the fused-node case. Namely, we expect there to be a few nodes with a high degree of connectivity versus most that have a low degree of connectivity. Because the graph-cutting algorithm the authors have devised appears to only normalize edge-weights connected to the (putatively) fused node "a", it seems miss the

opportunity of weeding out edges that belong to over-connected nodes (those at the “tail” of the degree distribution) for which most edges lie outside of the locally-defined sub-graph around “a”.

Response:

We thank the reviewer for their kind words. If we understand their point correctly, they may have slightly misunderstood what happens in the fused node correction algorithm. The edge weights between the two groups of connected nodes are themselves not normalized or affected. Rather, we use the graph partition and the resulting normalized cut as a parameter to determine whether a node should be split. If it is (given a cutoff), then the putatively fused node is removed, and two nodes are added each inheriting the connections to the nodes in either of the graph partitions. We have clarified the explanation further, hoping to resolve the confusion.

2. The authors compare filtering on indirect-connections versus read-counts and show an improvement in data retention. This is useful, but does potentially point to a work-around that it would be good to at least get some discussion on (if not some additional analysis). Ostensibly, constraining the inferred position solution to the top eigenvectors of the UEI/adjacency matrix would implicitly maximize the prominence of the highly connected portions of the data set. To the extent that the more “surgical” approach the authors take is better, one might wonder if the eigenvector calculation itself might improve with better normalization (and avoid the current apparent drawback of the authors' method requiring calculation on a much-less-sparse matrix than what is present in the raw data set). For example, as pointed out above, the long-tailed distribution of degrees-per-node might be having an undue influence on these data sets. De-weighting edges that connect to nodes of high non-local connectivity may have a similar effect as the authors' procedure (and would be computationally more efficient).

Response:

This raises an interesting point indeed, as the connections in highly-connected nodes could indeed be more prominent in the top eigenvectors. However, the reconstruction pipeline already takes this into account, by normalizing the UEI/adjacency matrix prior to the calculation of the eigenvectors. Likely this normalization was added due to naturally occurring biases in experimental data, that causes an uneven distribution of connectivity. We have tested whether this normalization is appropriate by applying a 2x connectivity bias to randomly selected nodes, and found that this did not affect the reconstruction quality in any of the simulations. We therefore feel confident in claiming that the distortion in inaccuracies arise out the crosslinks caused by the fused nodes themselves, and not by the presence of nodes with twice as many connections as others.

Reviewer #1 (Remarks on code availability):

I have reviewed the code but have not independently run it.

Reviewer #2 (Remarks to the Author):

The authors consider a novel DNA sequencing -based nonoptical microscopy technique, whereby individually barcoded short DNA strands are first randomly deposited onto a primer surface and then locally amplified to form labeled polymer colonies or "polonies". Crosslinked concatemers from nearby polonies then provide information about adjacencies of the seed-strand primer sites which can be used, post-sequencing, to reconstruct the spatial arrangement of the seed locations on the primer lawn.

The reconstruction can however be distorted by at least two types of experimental artefacts: (i) spurious crosslinks fabricated by incomplete PCR in the postprocessing phase and (ii) polonies which are fused together because of incidentally identical seed-strand barcodes, due to either coincidence or as a by-product of sequencing-error correction.

The manuscript presents computational methods for identifying and pruning these artefacts by: (i) validating all crosslinks by checking for the presence of matching short indirect connecting paths in the polony interconnection network and (ii) checking for the cohesion of polony neighborhoods by determining minimal-cost partitions for them. (The premise being that the network neighborhoods of artificially fused polonies would easily partition into components, whereas the network neighborhoods of true polonies would be densely connected.)

The methods are validated on a few numerical datasets and on one experimental dataset from an earlier publication. The numerical results look quite promising, and on the experimental dataset an error-correction quality comparable to earlier results is achieved with significantly reduced data loss, i.e. smaller number of pruned connections. (The captioning of the figures reporting the results could still be improved, though, particularly in the Extended Data section where it is not always clear which data sets are being considered and how the figures are to be interpreted.)

In general, I thus think that the manuscript addresses an essential concern in the development of a highly promising new technology in a tentatively promising manner.

However I was quite confused about the way the material is presented. The first Section of the "Results" chapter of the manuscript outlines the proposed error-correction methods in the toy model of a complete bipartite 2D grid graph, which is neither a good proxy for the actual polony interconnection networks, nor even close to the actual graph model used in the authors' own implementation of their methods, as discussed in the second Section of the "Results" chapter. (These issues are discussed more extensively in my attached "Detailed comments".)

The relationship of these two Sections is not explained to the reader, and in retrospect I actually find the whole presentation in terms of the 2D grid model superfluous and misleading. (Moreover this first Section contains a number of mathematical inconsistencies and inaccuracies which at points make following the argumentation there quite difficult; cf. again my attached "Detailed comments".)

I would thus propose that the authors carry out a fundamental rewrite of the manuscript, where they eliminate the confusing and misleading diversion via the 2D grid model and present their methods directly in terms of their actual diffusion-parameter based weighted-graph model. Such a rewrite would greatly improve the structure and readability of the manuscript, and as far as I can see is completely doable even following the general storyline of the present first Section but working in the framework of the actual weighted-graph model rather than the grid graph toy model.

Response:

We thank the reviewer for providing detailed comments and ideas (especially on the mathematical notation, which is not a domain in which we have a lot of experience). Regarding the perhaps strange organization, our intent was to first explain our algorithms in a simple setting before moving to the more complicated model, however, it appears to have caused more confusion than it might resolve. We have corrected the typos and reorganized the manuscript, removing the toy problem, moving the more formal description to the SI and starting straight away with the more realistic diffusion-based model.

We additionally address the comments given in the "detailed comments" section generally here. Each of these points has been clarified in the manuscript.

1. It is true that many of the strict derivations that apply for the toy problem do not apply for the diffusion-based model. Since we have removed the toy model, we have removed these statements, instead focusing on graded methods that are applicable in the diffusion-based model.
2. All the graphs on which the algorithm was applied are bipartite, undirected and weighted, both in the simulated data and experimental data
3. There is no strict guideline on what the best cutoff is in the diffusion-based dataset. However, in the simulated data, the algorithm does preferentially correct errors over true data, at any cutoff. We have included several guidelines how one might arrive at a good cutoff:
 - a. Using a quantile cutoff equal to the number of expected errors seems to work well in simulated data.
 - b. Using a too high cutoff does not seem to disrupt the reconstructions, unless a very high cutoff (50%) is used, suggesting a strict cutoff might not matter too much.
 - c. The effectiveness of the algorithm mostly depends on the spread: i.e, the wider the spread, the harder it becomes to distinguish errors from normal data (see ext. data 4 in the manuscript).

- d. Manually investigating the distribution of indirect path values or normalized cuts and finding a cutoff based could additionally be helpful
 In practice, these guidelines will require experimentation on real datasets.
4. We have added additional discussion how using longer short indirect paths might improve the algorithm's performance on sparse data.

More detailed commentary on specific comments (we could not copy the text so we refer to the line numbers the comment refers to).

“Line 133: This definition only gives the cut value on a specific partition”.

Response:

That is indeed the goal. The cut value and ncut value are used to evaluate the partition as obtained by the spectral graph partition algorithm. So we believe that the suggested equation is not required. The text has been clarified.

Reviewer #3 (Remarks to the Author):

Dear Authors,

I was pleasantly surprised that withing such a hot topic as DNA-microscopy such insightful additions could be made, and read your manuscript with great interest.

Overall, I am convinced that the proposed method is a highly valuable addition to the technique and will serve to broaden the applicability of the method in the face of less-than-perfect experimental conditions.

The only major remark I have is that the novelty and impact of the method are to some extent lost because of the unnecessarily complex introduction and derivation trough graph theory.

Therefore it is my recommendation that lines 76-157 are reworted, and moved to the appendix/SI, and in their place a short paragraph is put clearly outlining how the procedure works, with frequent reference to figure 2 and providing an intuitive understanding of the 2 problems and why they can be detected with the proposed metrics.

Response:

We thank the reviewer for their kind words and suggestions. As stated above, the manuscript has been reorganized. The toy problem is removed, and the algorithms are described more intuitively in the main text, with the more formal description being moved to the SI as suggested.

Furthermore, as a minor remark I would like a further explanation on the treatment of edge nodes (at the border of the cell) and if the method could be altered to stop potential artifacts at these nodes which naturally have fewer connections

Response:

Although the method seems to work well in its current state, it should indeed be possible to use a normalized or bias-corrected version of the indirect path values which should account for variation in node density, such as at the edge of the sample. We have included a short section on this in the discussion.

Reviewer #3 (Remarks on code availability):

It appears as though the code was all present, and a suitable readme file was included. However, due to time constraints I did not install the required packages or run the code

Decision Letter, first revision:

Date: 28th November 23 13:54:04
Last Sent: 28th November 23 13:54:04
Triggered By: Fernando Chirigati
From: fernando.chirigati@us.nature.com
To: bjorn.hogberg@ki.se
CC: computationalscience@nature.com
BCC: fernando.chirigati@us.nature.com
Subject: AIP Decision on Manuscript NATCOMPUTSCI-23-0819B
Message: Our ref: NATCOMPUTSCI-23-0819B

28th November 2023

Dear Dr. Högberg,

Thank you for submitting your revised manuscript "An error correction strategy for image reconstruction by DNA sequencing microscopy" (NATCOMPUTSCI-23-0819B). It has now been seen by the original referees and their comments are below. The reviewers find that the paper has improved in revision, and therefore we'll be happy in principle to publish it in Nature Computational Science, pending minor revisions to satisfy the referees' final requests and to comply with our editorial and formatting guidelines.

We are now performing detailed checks on your paper and will send you a checklist detailing our editorial and formatting requirements very soon, most likely today. Please do not upload the final materials and make any revisions until you receive this additional information from us.

TRANSPARENT PEER REVIEW

Nature Computational Science offers a transparent peer review option for original research manuscripts. We encourage increased transparency in peer review by publishing the reviewer comments, author rebuttal letters and editorial decision letters if the authors agree. Such peer review material is made available as a supplementary peer review file. **Please remember to choose, using the manuscript system, whether or not you want to participate in transparent peer review.**

Thank you again for your interest in Nature Computational Science. Please do not hesitate to contact me if you have any questions.

Best,
Fernando

--

Fernando Chirigati, PhD
Chief Editor, Nature Computational Science
Nature Portfolio

ORCID

Reviewer #1 (Remarks to the Author):

The authors have addressed my comments/questions sufficiently.

Reviewer #3 (Remarks to the Author):

The rewriting of the manuscript has greatly increased legibility. In its current form i feel that it will serve as a valuable resource for the community undoubtedly spurring on further developments.

Reviewer #3 (Remarks on code availability):

/

2nd review report on NATCOMPUTSCI-23-0819A, “An error correction strategy for image reconstruction by DNA sequencing microscopy”

Thank you for the opportunity to re-review this interesting manuscript. I think the presentation of the material has now been significantly improved, both in structure and in detail, and I am glad to recommend the manuscript for publication. I am suggesting below a number of small further corrections and improvements that I hope the authors can take into account when preparing their final version of the manuscript.

Detailed comments:

Line 79: 'setup introduced previously, that has yielded' → 'setup introduced previously that has yielded'

Lines 86–87: 'weights of the edges equals the number of products' → 'weights of the edges equal the observed number of products'? 'weights of the edges correspond to the number of products'?

Line 87: 'Seeing as two types' → 'Since two types'

Line 94: 'two polonies i and j , each of a different type' → 'two polonies i and j of different types'?

Lines 174–175: 'data simulated with a large spread' → 'simulated data with a large spread'?

Lines 186–187: This gap between typical distances covered by regular vs. spurious crosslinks is not that large, as can also be observed in Fig. 4B. (The averages seem to be well within one standard deviation of each other.) Is this a general characteristic of the method or a property of this specific dataset? Is this the (or one?) reason why spurious crosslink removal in the numerical experiments in general was not as successful as fused node splitting, and is there something that could be done about this issue?

Line 235 (Figure 2) and ff: I started wondering about the color scheme used in the figures here and later in the manuscript. It seems that typically dark color shades correspond to low values or disparity, overlap, fraction, etc., and light color shades correspond to high values. I notice that this goes against at least my na(t)ive interpretation of visuals and makes the figures always a bit unnecessarily hard to apprehend.

Line 247 (Figure 3): This is otherwise a very good and clarifying illustration, but the lowest diagram on fused node splitting seems to be missing the mark of the

“fused node” itself in the second panel. This seems like a little thing, but may create momentary confusion for a first-time reader, because one is here shifting the focus from a node a to its neighborhood S_a between panels 1 and 2 in both the upper (“normal”) and the lower (“fused”) diagram.

Lines 276–277: ‘used as edge weight’ → ‘used as the edge weight between the respective nodes’?

Line 286: ‘5D’ → ‘5E’

Line 291: ‘5E’ → ‘5D’

Line 330: ‘indirect paths at three steps’ → ‘indirect paths with three steps’, ‘indirect paths of length three’

Line 344: ‘Although MiniPath preferentially removes errors over real data, they are not completely separated’. ‘over’ → ‘in’? Also please clarify the referent of the pronoun ‘they’.

Line 346: ‘appropriate cutoff, that balances’ → ‘appropriate cutoff that balances’. Also the formulation ‘balances removal of real data with the removal of errors’ could possibly still be streamlined.

Line 349: ‘fraction of fused errors’ → ‘fraction of fusion errors’

Lines 350–351: ‘cutoff that is too high did not seem to negatively affect’ → ‘cutoff that is somewhat too high did not seem to negatively affect’

Line 366: ‘expected performance’ → ‘expected runtime performance’

Line 367: ‘ $O(d^k * |E|) = O(E)$ ’ → ‘ $O(d^k * |E|)$ ’. (Remove continuation ‘ $= O(E)$ ’ as the impact of d and k cannot in general be completely ignored.)

Line 367: ‘where E is the set of edges’ → ‘where $|E|$ is the number of edges in the connection graph’

Line 372: ‘applied onto any dataset’ → ‘applied on any dataset’

Line 379: ‘algorithm of adjacency data, that can detect’ → ‘algorithm of adjacency data that can detect’

Line 379: ‘implemented in python to calculate indirect paths’ → ‘implemented in Python to calculate weighted indirect paths’

Lines 386–388: 'starts from the asymmetric adjacency matrix [...], then calculating [...], then, for each node pair with an edge, subtracting the paths [...], while setting the other node pairs to 0' → 'starts from the asymmetric weight matrix [...], then calculates [...] and then, for each node pair with an edge, subtracts the weight of paths [...], while setting the other node pair weights to 0'

Lines 390–391: 'becomes computationally too challenging for large datasets, but since it calculates all paths of length three' → 'becomes computationally challenging for large datasets, partly because the method calculates all paths of length three'

Line 406: 'python' → 'Python'

Line 406: 'using spectral graph partitioning tool' → 'using spectral a graph partitioning tool'

Line 482: 'Machine Learning in {P}ython' → 'Machine Learning in Python'

Line 540 (Extended data 9): What are the values on the x and y axes here?

SI: Formal description [...] **Line 6:** ' U and V are the independent sets of nodes of the bipartite graph' → ' U and V are the independent sets nodes defining the bipartition of the graph'. (As the graph could have many other independent sets too.)

SI: Formal description [...] **Line 7:** 'assigning some edge weights d to each connected pair of nodes' → 'assigning an edge weight d to each connected pair of nodes'

SI: Formal description [...] **Lines 12–13:** 'taking the product of the edge weights [...], then adding this together for each indirect path found between them' → 'taking the product of the edge weights [...], and then adding these products together for all indirect paths connecting the nodes'

SI: Formal description [...] **Line 23:** ' E_S ' → ' E_a '

SI: Formal description [...] **Line 30:** 'To obtain the partition of G_a , we first check if it is' → 'To obtain the partition of G_a , we first check if G_a is'

SI: Formal description [...] **Lines 36–37:** 'edge weights removed between the S_{a1} and S_{a2} ' → 'edge weights removed between S_{a1} and S_{a2} ' (or 'edge weights removed between sets S_{a1} and S_{a2} ')

SI: Formal description [...] **Lines 42–43:** 'node a is removed two nodes are created in G that each inherit the edges to either the nodes in S_{a1} or S_{a2} ' → 'node a

is removed and two nodes are created in G that each inherit the edges to either the nodes in S_{a1} or the nodes in S_{a2} '

Final Decision Letter:

Date: 13th December 23 14:50:24

Last Sent: 13th December 23 14:50:24

Triggered By: Kaitlin McCardle

From: kaitlin.mccardle@us.nature.com

To: bjorn.hogberg@ki.se

BCC: kaitlin.mccardle@us.nature.com,computationalscience@nature.com,fernando.chirigati@us.nature.com,rjsproduction@springernature.com

Subject: Decision on Nature Computational Science manuscript NATCOMPUTSCI-23-0819C

Message Dear Professor Högberg,

:

We are pleased to inform you that your Article "An error correction strategy for image reconstruction by DNA sequencing microscopy" has now been accepted for publication in Nature Computational Science.

Once your manuscript is typeset, you will receive an email with a link to choose the appropriate publishing options for your paper and our Author Services team will be in touch regarding any additional information that may be required.

Please note that *Nature Computational Science* is a Transformative Journal (TJ). Authors may publish their research with us through the traditional subscription access route or make their paper immediately open access through payment of an article-processing charge (APC). Authors will not be required to make a final decision about access to their article until it has been accepted. [Find out more about Transformative Journals](https://www.springernature.com/gp/open-research/transformative-journals)

Acceptance of your manuscript is conditional on all authors' agreement with our publication policies (see <https://www.nature.com/natcomputsci/for-authors>). In particular your manuscript must not be published elsewhere and there must be no announcement of the work to any media outlet until the publication date (the day on which it is uploaded onto our web site).

Before your manuscript is typeset, we will edit the text to ensure it is intelligible to our wide readership and conforms to house style. We look particularly carefully at the titles of all papers to ensure that they are relatively brief and understandable.

Once your manuscript is typeset, you will receive a link to your electronic proof via email with a request to make any corrections within 48 hours. If, when you receive your proof, you cannot meet this deadline, please inform us at rjsproduction@springernature.com immediately.

If you have queries at any point during the production process then please contact the production team at rjsproduction@springernature.com.

We welcome the submission of potential cover material (including a short caption of around 40 words) related to your manuscript; suggestions should be sent to Nature Computational Science as electronic files (the image should be 300 dpi at 210 x 297 mm in either TIFF or JPEG format). We also welcome suggestions for the Hero Image, which appears at the top of our [home page](http://www.nature.com/natcomputsci); these should be 72 dpi at 1400 x 400 pixels in JPEG format. Please note that such pictures should be selected more for their aesthetic appeal than for their scientific content, and that colour images work better than black and white or grayscale images. Please do not try to design a cover with the Nature Computational Science logo etc., and please do not submit composites of images related to your work. I am sure you will understand that we cannot make any promise as to whether any of your suggestions might be selected for the cover of the journal.

You can now use a single sign-on for all your accounts, view the status of all your manuscript submissions and reviews, access usage statistics for your published articles

and download a record of your refereeing activity for the Nature journals.

Best regards,

Kaitlin McCardle, PhD
Senior Editor
Nature Computational Science

P.S. Click on the following link if you would like to recommend Nature Computational Science to your librarian: https://www.springernature.com/gp/librarians/recommend-to-your-library

** Visit the Springer Nature Editorial and Publishing website at www.springernature.com/editorial-and-publishing-jobs for more information about our career opportunities. If you have any questions please click here. **